# A calcium-based plasticity model for predicting long-term potentiation and depression in the neocortex

Giuseppe Chindemi [1✉], Marwan Abdellah[1], Oren Amsalem [2,3], Ruth Benavides-Piccione[4,5], Vincent Delattre[6], Michael Doron[7], András Ecker [1], Aurélien T. Jaquier[1], James King [1], Pramod Kumbhar [1], Caitlin Monney[1], Rodrigo Perin [6], Christian Rössert[1], Anil M. Tuncel [1], Werner Van Geit [1], Javier DeFelipe [4,5], Michael Graupner [8], Idan Segev[2,7], Henry Markram[1,6] & Eilif B. Muller [1,9,10,11✉]

Pyramidal cells (PCs) form the backbone of the layered structure of the neocortex, and plasticity of their synapses is thought to underlie learning in the brain. However, such long-term synaptic changes have been experimentally characterized between only a few types of PCs, posing a significant barrier for studying neocortical learning mechanisms. Here we introduce a model of synaptic plasticity based on data-constrained postsynaptic calcium dynamics, and show in a neocortical microcircuit model that a single parameter set is sufficient to unify the available experimental findings on long-term potentiation (LTP) and long-term depression (LTD) of PC connections. In particular, we find that the diverse plasticity outcomes across the different PC types can be explained by cell-type-specific synaptic physiology, cell morphology and innervation patterns, without requiring type-specific plasticity. Generalizing the model to in vivo extracellular calcium concentrations, we predict qualitatively different plasticity dynamics from those observed in vitro. This work provides a first comprehensive null model for LTP/LTD between neocortical PC types in vivo, and an open framework for further developing models of cortical synaptic plasticity.

[1] Blue Brain Project, École Polytechnique Fédérale de Lausanne, Geneva, Switzerland. [2] Department of Neurobiology, the Hebrew University of Jerusalem, Jerusalem, Israel. [3] Division of Endocrinology, Diabetes and Metabolism, Beth Israel Deaconess Medical Center, Harvard Medical School, Boston, MA 02215, USA. [4] Instituto Cajal, Consejo Superior de Investigaciones Científicas, Madrid, Spain. [5] Laboratorio Cajal de Circuitos Corticales, Centro de Tecnología Biomédica, Universidad Politécnica de Madrid, Madrid, Spain. [6] Laboratory of Neural Microcircuitry, École Polytechnique Fédérale de Lausanne, Lausanne, Switzerland. [7] Edmond and Lily Safra Center for Brain Sciences, the Hebrew University of Jerusalem, Jerusalem, Israel. [8] Université de Paris, SPPIN - Saints-Pères Paris Institute for the Neurosciences, CNRS, Paris, France. [9] Department of Neurosciences, Faculty of Medicine, Université de Montréal, Montréal, QC, Canada. [10] CHU Sainte-Justine Research Center, Montréal, QC, Canada. [11] Quebec Artificial Intelligence Institute (Mila), Montréal, Canada. ✉email: giuseppe.chindemi@unige.ch; eilif.muller@umontreal.ca

One of the most enigmatic features of the neocortex is the stereotypical layered architecture, which is thought to enable multiple interacting streams of information processing and play an important role in learning. PCs account for over 80% of the neurons in the neocortex and form the backbone of this laminar structure. Depending on the layer of origin, the extent of the dendritic tree, the axonal projections, and transcriptomic information[1], PCs can be subdivided into multiple classes with different input-output properties[1–6].

PCs interact with each other via a complex network of synaptic connections, which undergo persistent changes as a function of external stimuli and internal dynamics, thought to be the basis of learning[7]. This process is known as synaptic plasticity, and can manifest as LTP or LTD of synaptic efficacy[8–10].

Four decades of in vitro experiments have characterized the most important molecular pathways of LTP/LTD and found a rich set of stimulation protocols to explore the variables controlling plasticity induction (for a review see Malenka and Bear[11]). For example, spike-timing dependent plasticity (STDP) is a form of persistent synaptic efficacy modification where the direction and magnitude of synaptic changes are controlled by the relative timing of pre- and post-synaptic spikes[12–14].

Many models have been proposed for LTP/LTD of PC connections, ranging from phenomenological descriptions[15–22] to molecular level models of single synapses[23,24] (for reviews see Manninen et al.[25] and Kotaleski and Blackwell[26]). It is widely accepted that postsynaptic calcium, entering dendritic spines via N-methyl-D-aspartate receptors (NMDARs) and voltage-dependent calcium channels (VDCCs), is the key signal driving LTP/LTD[27–30]. The postsynaptic calcium transients can subsequently trigger the expression of plasticity either presynaptically, postsynaptically, or both[31–33]. The most common assumption on how calcium dynamics are linked to changes of synaptic efficacy is the so-called calcium-control hypothesis[27], whereby calcium transients of large amplitude are postulated to induce LTP, while prolonged calcium transients of moderate amplitude would result in LTD, but other forms considering multiple calcium sensors have been proposed[17,28,34].

Although the calcium-control hypothesis provides a strong foundation for describing plasticity mechanisms, the experimental coverage of LTP/LTD between PC connection types is still sparse. This poses a major barrier to the study of learning orchestration in the neocortex. Based on the currently available evidence, it is possible that every PC connection type implements a unique and specific "learning rule" within the context of the neocortical learning algorithm[35]. On the other hand, the in vitro reports on location-dependent synaptic plasticity indicate that connection-type specificity could be compatible with a uniform plasticity mechanism which is shaped by the physiological properties of the synaptic connection[36–38].

Given the critical role of calcium for the induction of synaptic plasticity, it is also important to consider that in vitro experiments are usually performed at an elevated extracellular calcium concentration. However, as synaptic plasticity depends crucially on the dynamics of neurotransmitter release and post-synaptic calcium influx, a non-physiological calcium concentration could produce plastic changes that are not representative of the true learning rules in vivo[39].

Here we use a modeling approach to evaluate the hypothesis that PC type-specific variability in post-synaptic calcium dynamics (driving plasticity) is sufficient to unify the available experimental findings on the heterogeneity of plasticity between PCs in the neocortex. First, we extended and calibrated a model of neocortical circuit[5] to explicitly describe postsynaptic calcium dynamics. Connected pairs of PCs were then sampled and simulated to mimic in vitro experiments on synaptic plasticity in

paired recordings[14,36,40–44]. These in silico experiments were used to adapt and optimize a previously described calcium-based model of LTP/LTD[19], which we then tested on held-out stimulation protocols and PC connection types. Finally, we used the model to predict plasticity outcomes under physiological extracellular calcium conditions.

## Results

**Modeling synaptic plasticity of PC connections.** To test our hypothesis that calcium signaling diversity can explain plasticity diversity, we introduced calcium-dependent synaptic plasticity into a previously developed model of a neocortical microcircuit[5,45] (Fig. 1a). The circuit model comprises 163271 compartmental neurons, based on morphological reconstructions of 17 pyramidal cell types across 6 layers in the rat somatosensory cortex, and has been extensively validated in terms of synapse location on the dendrites, number of synapses per connection, in vivo spontaneous and stimulus-evoked dynamics, and stochasticity and multi-vesicular nature of neurotransmission[5,46–48]. We extracted and simulated pairs of connected PCs from the circuit model to obtain a representative sample for each connection type (i.e. 100 connections each). Importantly, sampling connected pairs from a larger microcircuit model allowed us to transparently account for the variability of morphologies, innervation profiles and synaptic transmission parameters within each connection type. All in silico models and simulations used in this work are made publicly available at https://doi.org/10.5281/zenodo.5654788[49].

For PC connections, the interplay of pre- and postsynaptic activity causes calcium influx into the spine head due to NMDARs and VDCCs, driving the induction of synaptic plasticity (Fig. 1b). We estimated the fractional calcium current due to NMDARs by extending the approach of Jahr and Stevens[50], and fitting to recent data for the neocortex[51,52]. VDCCs were modeled as an inactivating population of R-type channels in the Hodgkin–Huxley (HH) formalism and calibrated to experimental data[53,54]. Spine calcium dynamics were described using a point current formalism with instantaneous buffering[55,56], resulting in a linear ordinary differential equation (ODE) as follows:

$$\frac{d}{dt}[\mathrm{Ca}^{2+}]_\mathrm{i} = (\tilde{I}_\mathrm{NMDAR} + I_\mathrm{VDCC})\frac{\eta}{2F \cdot X} - \frac{\left([\mathrm{Ca}^{2+}]_\mathrm{i} - [\mathrm{Ca}^{2+}]_\mathrm{i}^{(0)}\right)}{\tau_\mathrm{Ca}},$$

(1)

where $[\mathrm{Ca}^{2+}]_\mathrm{i}$ is the free calcium concentration in the spine head, $\tilde{I}_\mathrm{NMDAR}$ is the calcium component of the NMDAR-mediated current, $I_\mathrm{VDCC}$ is the VDCC-mediated calcium current, $\eta$ is the fraction of free (non buffered) calcium, $F$ is the Faraday constant, $X$ is the spine volume, $[\mathrm{Ca}^{2+}]_\mathrm{i}^{(0)}$ is the intracellular calcium concentration at rest, and $\tau_\mathrm{Ca}$ is the time constant of free calcium clearance.

To accurately model the distribution of calcium transients, it is important to consider the known correlation between spine volume and calcium conductances, as can be inferred from experiments[57–59], and assign large and small spine volumes to high- and low-conductance synapses, respectively. Accounting for this correlation has the effect of reducing calcium transient variability across synapses. To this end, we used the total synaptic conductance, as prescribed by constraints of the circuit model, to determine the spine volume, $X$ (see Methods). Spine volume is then used to estimate the VDCC conductance, assuming a spherical spine head and using surface density measurements reported in literature[54].

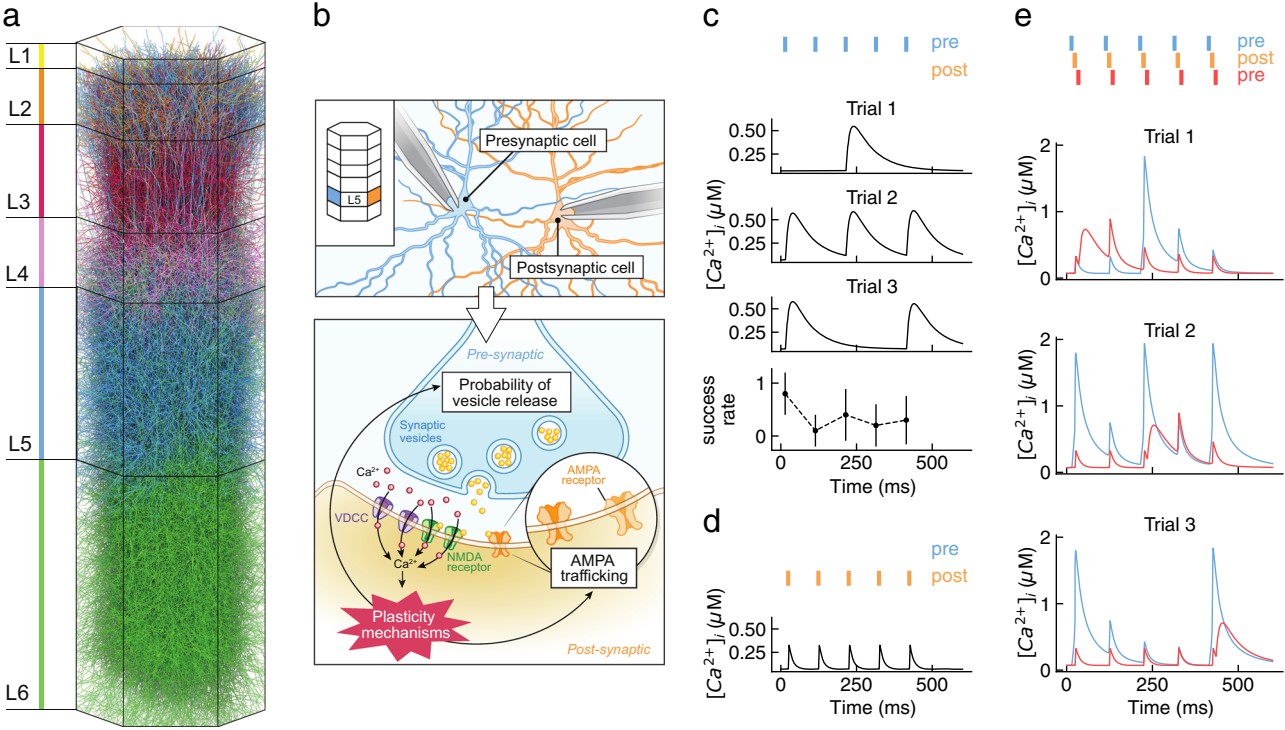

**Fig. 1 Modeling the role of postsynaptic calcium in LTP/LTD. a** A 3-D rendering of a neocortical column in the circuit model used in this study. Neuronal morphologies are colored according to their layer of origin (axons not shown). **b** Schematic of the main players and events contributing to LTP/LTD induction at excitatory synapses (bottom) between pyramidal neurons (top). Pre-synaptic vesicle release and subsequent post-synaptic depolarization results in calcium influx in dendritic spines via NMDARs and VDCCs. Calcium signaling activates independent biochemical pathways, leading to long-term changes in AMPAR conductance (postsynaptic expression mechanisms) and/or vesicle release probability (presynaptic expression mechanisms). **c** Calcium transients at a single synapse during repetitive firing of the presynaptic neuron at 10 Hz (upper three panels; Trial 1-3). Trial-to-trial variability is governed by the rapid depletion of the number of vesicles available for release and the resulting drop in success rate (bottom; data shown as mean ± STD, $n = 10$). **d** Calcium transients at a single synapse during repetitive firing of the postsynaptic neuron at 10 Hz. VDCCs respond with reliable calcium transients to postsynaptic spikes. **e** Calcium transients at a single synapse during repetitive pairing of the pre- and post-synaptic neuron at 10 Hz. Presynaptic spikes shortly preceding postsynaptic spikes (blue, +10 ms) cause nonlinear activation of the NMDARs and a large calcium influx. The inverse timing relationship (post before pre, red, −10 ms) results in an almost linear summation of NMDAR- and VDCC-mediated calcium transients.

Within this formalism, pre- or post-synaptic spiking results in a rich repertoire of calcium transients due to the complex interactions between the calcium current sources described above and the neurotransmission dynamics prescribed by the circuit model[5,48]. Isolated presynaptic spikes primarily trigger NMDAR-mediated calcium influx, but in a probabilistic manner governed by the vesicle release probability. We compared the calcium transients generated by our model for synaptic activation with those obtained experimentally in vitro by Sabatini et al.[56] and found good agreement for the mean peaks ($0.67 \pm 0.44\,\mu M$ in silico, $0.7 \pm 0.4\,\mu M$ in vitro; mean ± STD) and the timecourse (Supplementary Fig. A.1a–c). It should be noted that, as the timecourse of these calcium transients is determined by the slow NMDAR current, minor quantitative differences between the model and the in vitro data are expected, considering that the latter experiments were performed in the hippocampus. Repetitive activation of the presynaptic terminal, as commonly used to induce plasticity in vitro, rapidly depletes the small readily-releasable pool (RRP) of synaptic vesicles ($N_{RRP} \approx 2$[48,60,61]), thereby reducing the probability of inducing a postsynaptic calcium response (Fig. 1c). Conversely, isolated postsynaptic spikes only activate VDCCs, inducing reliable calcium influx even during repetitive firing (Fig. 1d). We compared the calcium transients generated by our model by a single postsynaptic spike with those obtained experimentally in vitro by Sabatini et al.[56] and found good agreement for the mean peaks ($1.4 \pm 0.6\,\mu M$ in silico, $1.7 \pm 0.6\,\mu M$ in vitro; mean ± STD) and the timecourse

under reasonable assumptions of the path length sampling distribution on which this quantity strongly depends (Supplementary Fig. A.1d–i). Importantly, the combination of these two calcium activation modalities can produce a cooperative non-linear interaction due to the sensitivity of NMDARs to membrane depolarization. That is, spiking-induced postsynaptic depolarization immediately following the activation of NMDARs (pre before post; blue; Fig. 1e) causes larger calcium transients than the opposite timing relationship (post before pre; red; Fig. 1e). The timing dependence of this nonlinear effect is thought to be the origin of STDP (for a review see Graupner and Brunel[29]), a well-studied component of PC plasticity in the neocortex.

To account for the frequency dependence of LTP[12], the calcium control hypothesis implicitly presupposes supralinear calcium accumulation in the spine during high-frequency stimulation[27] (for a review, see Manninen et al.[25]). Our calcium model, which was constrained by experimental measurements of calcium dynamics in spines, predicts that such accumulation is not produced at cortical PCs by traditional high-frequency stimulation protocols (Fig. 2a top). This can be attributed to the stochasticity of vesicle release, the low number of vesicles in the RRP of individual synapses[48], and the fast calcium clearance mechanisms[56]. Consistent with this, previous theoretical work has identified the importance of introducing longer calcium integration time constants to explain plasticity outcomes[17]. Following this work, we introduced a leaky calcium integrator, $c^{\star}$, to capture the frequency dependence of LTP while

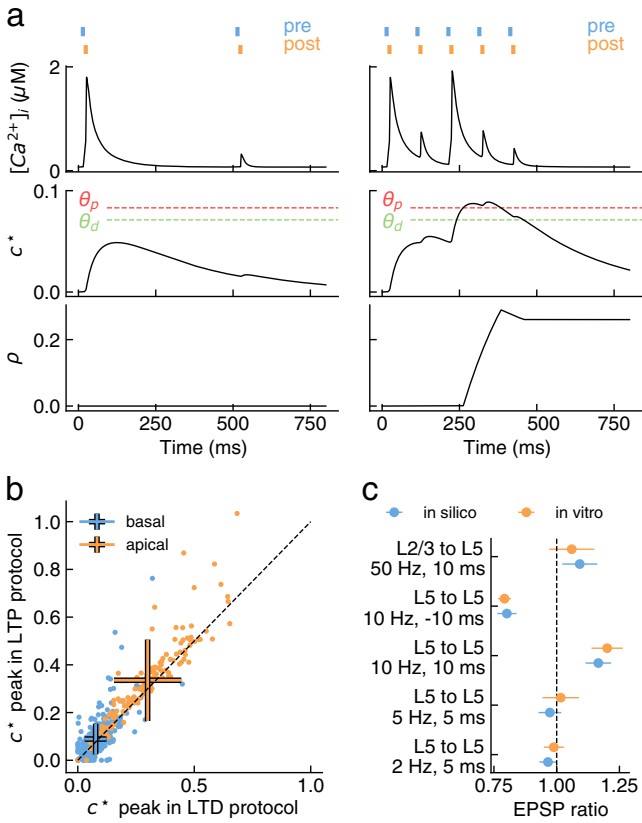

**Fig. 2 Plasticity model components and parameter optimization.**
**a** Evolution of plasticity model state variables during coincident activation of pre- and postsynaptic neurons at 2 Hz (left) and at 10 Hz (right). The instantaneous calcium concentration ($c$; top) is integrated overtime to produce a suitable readout ($c^\star$; middle) for plastic changes, expressed as an increase of synaptic efficacy ($\rho$; bottom). Crossing the depression threshold ($\theta_d$; green) causes a decrease in the synaptic efficacy whereas crossing the potentiation threshold ($\theta_p$; red) results in an increase. **b** Comparison of calcium integrator peaks during LTD-inducing protocol (10 Hz, $-10$ ms) against LTP-inducing protocol (10 Hz, $+10$ ms). Peaks are well clustered by dendritic location (apical, orange; basal, blue) and on average slightly larger during the LTP protocol then during the LTD protocol. Error bars represent standard deviation and intersect at the mean value of the cluster. Peaks were estimated from a single repetition of the stimulation protocol at each synapse. **c** Comparison of mean EPSP ratio between model (in silico, blue) and experimental constraints (in vitro, orange) after model parameter optimization. Experimental data (in vitro) from Markram et al.[12], Sjöström and Häusser[36] (see Table 1). Data reported as mean ± SEM. Welch's unequal variances two-sided t-test was n.s. for every protocol (p-value from top to bottom: 0.78, 0.84, 0.67, 0.63, 0.69; $n = 100$).

maintaining a simple thresholding plasticity mechanism, as follows

$$\frac{d}{dt}c^\star = -\frac{c^\star}{\tau_\star} + \left([Ca^{2+}]_i - [Ca^{2+}]_i^{(0)}\right), \tag{2}$$

where $\tau_\star$ is the integration time constant. This mechanism integrates calcium over longer timescales to drive plasticity, as could arise from the interplay of fast and a slow buffers in the spine head[24,62], involving for example calmodulin[56]. Functionally, it gives heavier weight to longer NMDAR-mediated over shorter VDCC-mediated calcium events, and to calcium events arriving in close proximity, such as during high frequency stimulation (Fig. 2a middle).

Expression of synaptic plasticity takes the form of persistent changes to synaptic parameters, which we assume here to be driven by the integrated calcium, $c^\star$. To model these persistent changes, we followed the approach of Graupner and Brunel[19]. In this formalism, the "synaptic efficacy" is a dynamic variable, $\rho$, driven by the integrated calcium concentration, and exhibiting bistable dynamics (see Fig. 2a bottom), according to the following equation

$$\frac{d}{dt}\rho = \left(-\rho(1-\rho)(0.5-\rho) + \gamma_p(1-\rho)\Theta[c^\star - \theta_p] - \gamma_d\rho\Theta[c^\star - \theta_d]\right)/\tau, \tag{3}$$

where $\tau$ is the time constant of convergence of the synaptic efficacy, $\rho = 0.5$ is the unstable fixed point separating the basins of attraction of the two stable states (depressed at $\rho = 0$ and potentiated at $\rho = 1$), $\Theta$ is the Heaviside function, $\theta_d$ and $\theta_p$ are the depression and potentiation thresholds, and $\gamma_d$ and $\gamma_p$ are the depression and potentiation rates, respectively.

Plasticity of neocortical synapses between pyramidal cells has been found to be expressed presynaptically as a persistent increase/decrease of vesicle release probability[31,63], and postsynaptically as alpha-amino-3-hydroxy-5-methyl-4-isoxazole propionate receptor (AMPAR) insertion/removal[32]. Furthermore, the expression mechanisms are slower than induction mechanisms, as demonstrated by the slow buildup of LTP/LTD commonly observed after plasticity-induction protocols in vitro[12,14]. To account for all these effects, the synaptic efficacy $\rho$ is dynamically converted into a release probability, $U_{SE}$, and AMPAR conductance, $\hat{G}_{AMPAR}$ by low-pass filtering as follows

$$\frac{d}{dt}U_{SE} = \frac{\bar{U}_{SE} - U_{SE}}{\tau_{change}}$$
$$\bar{U}_{SE} = U_{SE}^{(d)} + \rho(t) \cdot \left(U_{SE}^{(p)} - U_{SE}^{(d)}\right) \tag{4}$$

$$\frac{d}{dt}\hat{G}_{AMPAR} = \frac{\bar{G}_{AMPAR} - \hat{G}_{AMPAR}}{\tau_{change}}$$
$$\bar{G}_{AMPAR} = \hat{G}_{AMPAR}^{(d)} + \rho(t) \cdot \left(\hat{G}_{AMPAR}^{(p)} - \hat{G}_{AMPAR}^{(d)}\right), \tag{5}$$

where $U_{SE}^{(d)}, U_{SE}^{(p)}, \hat{G}_{AMPAR}^{(d)}, \hat{G}_{AMPAR}^{(p)}$ are constants parameterizing a linear conversion of the depressed (d) and potentiated (p) states to release probability $U_{SE}$ and AMPAR conductance $\hat{G}_{AMPAR}$. For simplicity we assumed that these two synaptic variables evolve together by assigning the filtering time constants to be identical ($\tau_{change}$). Moreover, by using $\rho$ to drive both pre-synaptic and post-synaptic changes, we implicitly assumed that these two synaptic variables are correlated, a view supported by experimental observations on the structural and functional relationships between synaptic variables[57–59,61,64].

While most parameters of the plasticity model can be extracted from literature or fixed to reasonable values, the available experimental data is insufficient to directly estimate the time constant of the calcium integrator ($\tau_\star$), the depression and potentiation thresholds ($\theta_d$, $\theta_p$) and the depression and potentiation rates ($\gamma_d$, $\gamma_p$). To determine these remaining parameters of the plasticity model, we replicated previous in vitro experiments using our chosen circuit model[5], and used optimization approaches to achieve a best match to the experimental data.

Specifically, we assumed that $\tau_\star$, $\gamma_d$ and $\gamma_p$ are shared across all synapses, and hypothesized that the thresholds for the induction of LTP/LTD are homeostatically regulated locally at each synapse based on the calcium amplitude observed during normal network activity (i.e. excitatory post-synaptic potentials (EPSPs) and backpropagating action potentials (bAPs)). That is, the thresholds were expressed as a linear combination of the calcium peak

**Table 1 Datasets and stimulation protocols from in vitro experiments.**

| Pre m-type | Post m-type | Area | Species | Age | Freq. (Hz) | Δt (ms) | Reference |
|---|---|---|---|---|---|---|---|
| L5-TTPC | L5-TTPC | SSC | rat | P14–P16 | 2[†], 5[†], 10, 20, 30, 40 | 5 | 12 |
| L5-TTPC | L5-TTPC | SSC | rat | P14–P16 | 10 | 10[†], −10[†] | 12 |
| L2/3-PC | L2/3-PC | SSC (barrel) | rat | P12–P14 | 20 | 10 | 41 |
| L4-SSC | L4-SSC | SSC (barrel) | rat | P12–P14 | 1, 10, 20, 50 | 10 | 41 |
| L5-TTPC | L5-TTPC | VC | rat | P12–P21 | 0.1, 10, 20, 40, 50 | 10, −10 | 14 |
| L5-TTPC | L5-TTPC | VC | rat | P12–P21 | 0.1, 20, 40, 50, 100 | 0 | 14 |
| L5-TTPC | L5-TTPC | VC | rat | P12–P21 | 20 | 25, −25 | 14 |
| L4-PC | L2/3-PC | SSC (barrel) | mouse | P9–P14 | 0.2 | 10, −15 | 42 |
| L2/3-PC | L2/3-PC | SSC (barrel) | mouse | P12–P18 | 0.2 | −15 | 43 |
| L2/3-PC | L2/3-PC | VC | rat | P14–P14 | 0.2 | 10, −10 | 44 |
| L2/3-PC | L2/3-PC | VC | rat | P14–P21 | 10 | 10 | 44 |
| L2/3-PC | L2/3-PC | VC | rat | P14–P21 | 20 | 10, −10 | 44 |
| L5-TTPC | L5-TTPC | VC | rat | P14–P21 | 50 | 10 | 36 |
| L2/3-PC | L5-TTPC | VC | rat | P14–P21 | 50[†] | 10 | 36 |

[†] Stimulation protocol used for model fitting.

during a single EPSP due to the release of all vesicles in the RRP, $C_{pre}$, and the calcium peak during a single bAP, $C_{post}$, generalizing the approach taken in Graupner and Brunel[19]. As these events occur much more often than plasticity-inducing pairings, frequently in isolation, and activate both NMDARs and VDCCs, they are ideal candidates to provide a reference point for the homeostatic regulation of the LTP/LTD thresholds. As calcium transients span a large range of amplitudes and form two clusters representing synapses on basal or apical dendrites (Fig. 2b), the parameters of the linear combinations were assumed to be shared between all basal and all apical synapses respectively, resulting in 8 threshold parameters to be optimized during model fitting (11 free parameters in total: the 8 threshold parameters, $\gamma_p$, $\gamma_d$, and $\tau_\star$; see Methods).

For optimization targets and testing comparisons, we considered all available datasets from previously published plasticity experiments for glutamatergic connections in the juvenile rodent neocortex that used whole-cell paired recordings: layer 5 thick-tufted pyramidal cell (L5-TTPC) to L5-TTPC[12,14], layer 2/3 pyramidal cell (L2/3-PC) to L5-TTPC[36], L2/3-PC to L2/3-PC[41,43,44], layer 4 pyramidal cell (L4-PC) to L2/3-PC[42], layer 4 spiny-stellate cell (L4-SSC) to L4-SSC[41] (see Table 1). Some of these datasets are potentially incompatible with the cellular and synaptic physiology of the circuit model presented here (rat somatosensory cortex), but were included anyway for an exhaustive comparison of all available rodent neocortical plasticity datasets using whole-cell paired recordings. We did not consider experiments performed in other brain regions, species, or developmental stages to be consistent with the underlying circuit model, which provides synaptic connectivity and physiology on which plasticity outcomes depend.

We used only the L5-TTPC to L5-TTPC[12] and L2/3-PC to L5-TTPC[36] datasets during model fitting (training set), holding out the other datasets to subsequently assess the predictive power of the model (test set). During model fitting, 200 pairs of connected PCs were randomly sampled from the circuit model (100 for each connection type, see Methods), and stimulated to pair the activity of the pre- and postsynaptic neurons at frequencies and time offsets matching the in vitro experiments. The mean EPSP amplitude was assessed during the minutes preceding the induction protocol and then monitored for 40 minutes after the induction. The last 60 EPSPs were used to assess the amplitude change with respect to baseline. We then used a multi-objective evolutionary strategy[65,66] to optimize the 11 free plasticity model parameters. Out of 10360 evaluated solutions, we found 83 valid candidates (i.e. errors within one SEM from the in vitro target for

each stimulation protocol; see Supplementary Fig. A.2). The best solution was then chosen as the one minimizing the maximum of its errors across all stimulation protocols in the training set. That is, a solution was considered as good as its largest error on the set of training protocols. Interestingly, the parameter sets prescribed by the valid candidates were clustered around the chosen best solution and their performance gradually degraded towards the perimeter of the cluster (Supplementary Fig. A.3). This result suggests that the optimization algorithm converged to a smooth local minimum and demonstrates a certain robustness of the best solution, as small variations of its parameters do not cause a catastrophic drop in performance on the training set. After optimization, the model captured the in vitro results for L5-TTPC to L5-TTPC and L2/3-PC to L5-TTPC plasticity (Fig. 2c; differences between the model and in vitro experiments are n.s.; Welch's unequal variances t-test).

Taken together, these results indicate that we do not need to optimize the free parameters for each synapse individually, but rather we can model the observed plasticity mechanisms with a shared set of free parameters. This suggests that the model could generalize to novel protocols, and predict plasticity outcomes at connections types that have not been experimentally character-ized. We tested this hypothesis, first on novel protocols for the optimized connection types (L5-TTPC to L5-TTPC and L2/3-PC to L5-TTPC), and subsequently on novel connection types without further parameter tuning, as described in the following two sections.

**Model validation and predictions on the optimized connection types.** We validated the model using novel stimulation protocols within the same classes of neurons used for model optimization. To ensure the plasticity model is not overfitting a specific sample of in silico PC connections, validation was performed on a novel set of connections drawn from the tissue model.

First, we validated the L5-TTPC to L5-TTPC connection type (see Supplementary Fig. A.23 for distributions of synaptic parameters). We extended the set of experimental protocols in Markram et al.[12] (Fig. 3a, b) to obtain a dense map of LTP/LTD dynamics for this connection type, as a function of stimulation frequency and time delay between the spike trains of the pre- and post-synaptic neurons (Fig. 3c). The cross-sections of this map allow the comparison of the in vitro observations[12] with the predictions of the in silico model. The model correctly reproduces the experimental LTP-frequency dependence, even though frequencies above 10 Hz were not included in the training set (Fig. 3d). Similar results were also

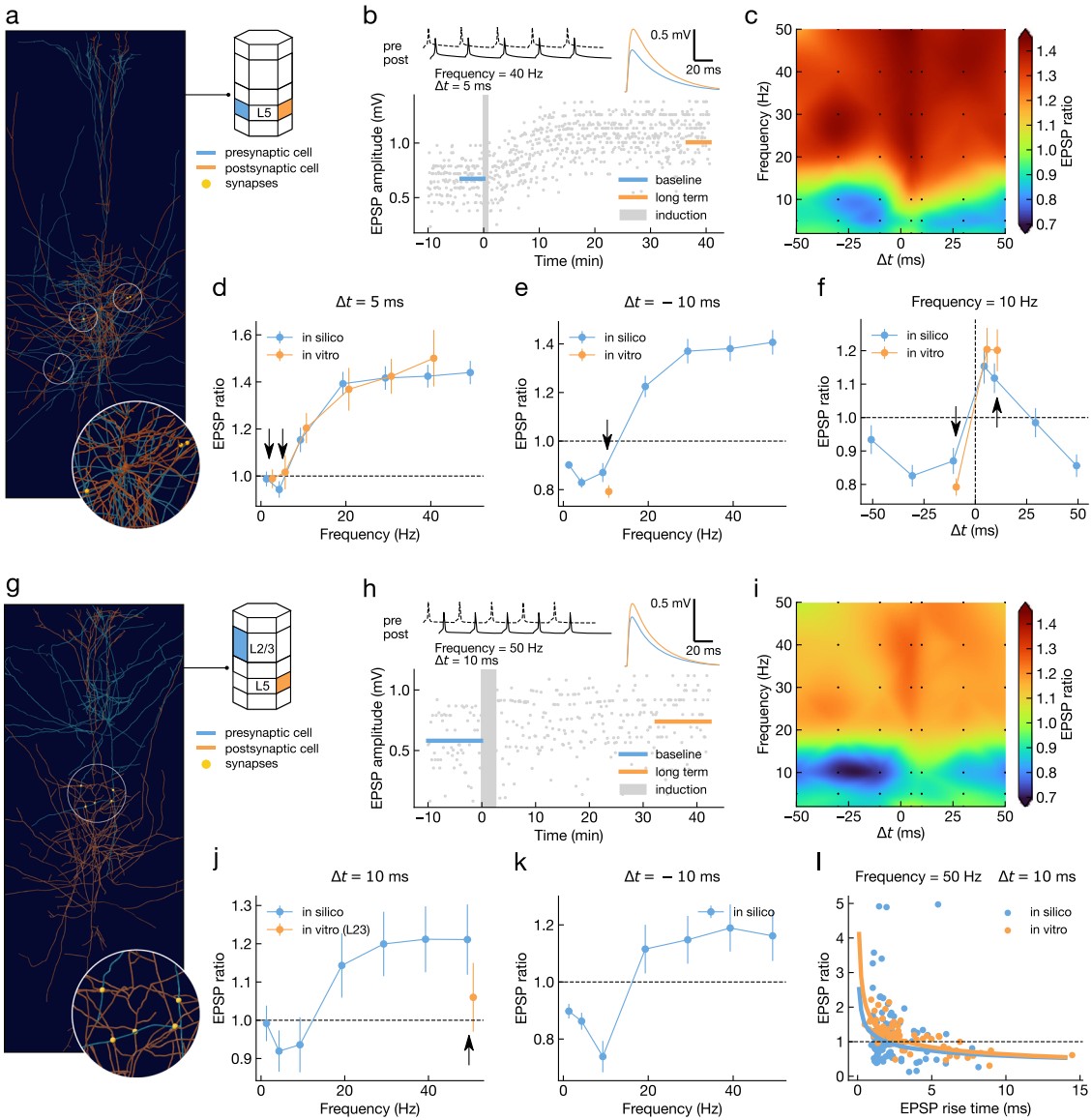

**Fig. 3 Comparison of in vitro and in silico LTP/LTD outcomes for two distinct populations of synapses onto L5-TTPC. a** 3-D rendering of a representative pair of connected L5-TTPCs in the in silico model. Inset shows a magnified view of some of the synapses mediating the connection (yellow spheres). **b** Evolution over time of simulated EPSP amplitude (bottom) during a typical plasticity induction protocol (top left; one burst shown out of 10). Mean EPSP amplitudes (top right) are shown before (baseline; blue) and after (long term; orange) the induction protocol. **c** Timing and frequency dependence of LTP/LTD in silico for the protocol described in **b** ($n = 100$). Simulated configurations shown as black markers, with cubic interpolation elsewhere. **d** Comparison of in vitro and in silico frequency dependence of synaptic changes at $\Delta t = +5$ ms. Welch's unequal variances two-sided t-test was n.s. for every protocol (*p-value* from low to high stimulation frequency: 0.988, 0.399, 0.550, 0.821, 0.940, 0.586; $n = 100$). **e** As in **d** for $\Delta t = -10$ ms. Welch's unequal variances two-sided t-test was n.s. (*p-value*: 0.103; $n = 100$). **f** Comparison of in silico and in vitro STDP at a frequency of 10 Hz. Welch's unequal variances two-sided t-test was n.s. for every protocol (*p-value* from negative to positive stimulation timing: 0.103, 0.550, 0.299; $n = 100$). Experimental data (in vitro) from Markram et al.[12]. Panels **g**–**k** as in **a**–**e** for L2/3-PC to L5-TTPC connections. For **j**, Welch's unequal variances two-sided t-test was n.s. (*p-value*: 0.245; $n = 100$). **l** Dependence of LTP/LTD on dendritic location. EPSP rise time is used as a correlate of the average contact distance from soma for comparison with Sjöström and Häusser[36] (in vitro; includes also connections between L5-TTPCs and those acquired via extracellular presynaptic stimulation). Circles represent individual experiments, while solid lines are power-law fits to the data (in silico, blue, $f(x) = 1.22x^{-0.32}$; in vitro, orange, $f(x) = 1.61x^{-0.41}$). All panels report EPSP ratio data as mean ± SEM. Black arrows indicate stimulation protocols used to optimize model parameters. For the full distribution of in silico experiment outcomes see Supplementary Fig. A.10 (L5-TTPC to L5-TTPC) and Supplementary Fig. A.11 (L2/3-PC to L5-TTPC).

observed in the visual cortex using a comparable experimental protocol[14], and were reproduced by our model (Supplementary Fig. A.4a). The observed quantitative match between in vitro and in silico data at higher frequencies is compatible with our assumptions on the initial ratio of potentiated over depressed synapses, and the conductance and release probabilities of the

depressed and potentiated states. That is, the maximum magnitude of LTP/LTD obtainable in our model is constrained by the initial state of all synapses (i.e potentiated or depressed) and by how much each synapse can change. For example, a pool of connections having all synapses initialized in the potentiated state will not produce LTP for any stimulation protocol, as there

are no synapses to potentiate. Similarly, if on average the conductance (release probability) of the potentiated state is only marginally larger than the one of the depressed state, even potentiating every synapse might not be enough to match the target in vitro experiments. These parameters were estimated before fitting, based on the available experimental evidence and theoretical considerations.

Next, we considered LTD-inducing protocols. The model correctly reproduces the experimental data (Fig. 3e), although a slightly better performance was obtained during training (see Fig. 2c), indicating the onset of neuron-sampling specific overfitting. Over the range of frequencies considered, we observed the emergence of LTD-frequency dependence, consistent with experiments in visual cortical slices[14] (Supplementary Fig. A.4b, c). Specifically, LTD is abolished at high-frequency stimulation for this connection type (see Fig. 3c). Interestingly, in our model the switch from LTD to LTP occurs at a lower stimulation frequency compared to in vitro data from visual cortex[14] (e.g. approx. 20 Hz vs approx. 40 Hz for $\Delta t = -10$ ms; see Supplementary Fig. A.4b). Since LTD in visual cortex also has a larger magnitude across protocols when compared to both our model and the in vitro data[12], these findings could indicate a tendency for greater depression in visual cortex compared to somatosensory cortex.

Our model assumes a single synaptic efficacy variable $\rho$, which is then expressed as pre- and post-synaptic changes. This assumption seems to contradict previous reports that the high-frequency stimulation protocols reproduced in this work can also induce pre- and post-synaptic changes of opposite directions (i.e. presynaptic LTD and postsynaptic LTP)[32]. However, the analysis approach to determine the locus of expression of synaptic plasticity only considers the net EPSP change measured at the soma, whereas the underlying local plastic changes across the multiple individual synapses mediating a given connection could have diverse and even contrasting directions and magnitudes. In this scenario, each of the multiple synaptic contacts might still undergo matching pre- and post-synaptic plasticity (as assumed here), and yet we could observe divergent pre- and post-synaptic LTP/LTD of the somatic EPSP. To test this hypothesis, we applied somatic analysis of the locus of expression, as in Sjöström et al.[32], to our simulation results and found that presynaptic LTD and postsynaptic LTP of the whole connection could still coexist in our model under the assumption of matched pre and post changes of individual synapses (Supplementary Fig. A.5a; data points in red shaded area). This indicates that reports of divergent pre- and post-synaptic LTP/LTD at the level of whole connections do not exclude a priori our assumption of a single synaptic efficacy variable. However, Sjöström et al.[32] also reported a lack of postsynaptic LTD in their experiments, which our model cannot account for (Fig. A.5b, c).

Finally, our model shows a clear STDP window at 10 Hz (Fig. 3f), also qualitatively consistent with experiments in visual cortical slices[14] (Supplementary Fig. A.4ad-h). Here we restricted our analysis to relative spike timings in the range of $-50$ to $+50$ ms, as traditionally considered in vitro, although our model could support longer integration windows due to the large time constant of the calcium integrator. Similar to what was observed for the LTD frequency dependence, STDP in our model is most evident at lower frequencies than reported in the visual cortex and completely abolished above 20 Hz. Furthermore, our model predicts a flip from LTP to LTD around $\Delta t = +50$ ms, a configuration where the pre- and post-synaptic spike trains are perfectly interleaved (i.e. the inter-spike intervals (ISIs) of the pre- and post-synaptic spike trains are $2 \times \Delta t$). This finding is in agreement with experiments in the visual cortex, where a flip from LTP to LTD was observed for $\Delta t = +25$ ms for a stimulation frequency of 20 Hz[14], a configuration where the pre- and post-

synaptic spike trains are also perfectly interleaved (Supplementary Fig. A.4f, orange).

Taken together, these results indicate quantitative agreement of the model with L5-TTPC to L5-TTPC plasticity dynamics in the somatosensory cortex and qualitative agreement with those in the visual cortex. Considering that our circuit model is specific for the somatosensory cortex, it might be possible that the same plasticity model parameterization determined here would provide a quantitative agreement for the visual cortex, given an accurate circuit model of that region. However, we cannot rule out the possibility of a region-specific parameterization being required.

We next validated the L2/3-PC to L5-TTPC connection type (Fig. 3g–i, see Supplementary Fig. A.24 for synaptic parameters distributions). Sjöström and Häusser[36] showed that a stimulation protocol causing large LTP in L5-TTPC to L5-TTPC connections would not potentiate L2/3-PC to L5-TTPC connections, and often produce depression instead. This effect is due to the distal location of L2/3-PC to L5-TTPCs synapses and the consequent attenuation of the bAPs involved in plasticity induction[36]. Our model reproduced this result (Fig. 3j), although a slightly better performance was obtained during training (see Fig. 1f), again indicating the onset of neuron-sampling specific overfitting. Considering a larger set of stimulation frequencies than is available from in vitro recordings revealed a form of frequency-dependent LTP/LTD for this connection type (Fig. 3j-k). In contrast to what was observed for L5-TTPC to L5-TTPC connections (see Fig. 3d, e), strong to moderate LTD was always present between 5 and 10 Hz. Furthermore, our model qualitatively reproduces distance-dependent plasticity (Fig. 3l), even though no requirements on distance-dependent effects were imposed during model fitting. Quantitative differences between in silico and in vitro data (KDE test, p-value < 0.001) could be due to the heterogeneity of the in vitro dataset (which includes extracellular recordings and L5-TTPC to L5-TTPC connections) or to cortical region discrepancies (somatosensory for in silico, visual for in vitro). However, power-law fits ($y = b \cdot x^a$) revealed statistically indistinguishable exponents ($a = -0.32 \pm 0.13$ in silico, $a = -0.41 \pm 0.04$ in vitro), suggesting that the two distributions exhibit an equivalent distance dependence up to a scaling factor ($b = 1.22 \pm 0.06$ in silico, $b = 1.61 \pm 0.04$ in vitro).

**Model testing and predictions on novel connection types.** We evaluated the generalization of the model using two additional classes of PCs not used during parameter optimization: L2/3-PC to L2/3-PC (Fig. 4) and L4-PC to L2/3-PC (Fig. 5).

First, we considered LTP between L2/3-PC connections due to pairing at 20 Hz stimulation[41] and found that our model qualitatively reproduces the in vitro result (Fig. 4b–d; see Supplementary Fig. A.26 for synaptic parameters distributions). Note that in the analysis we excluded one outlier connection in the in silico dataset, having a disproportionate skewing effect on the average EPSP ratio due to its small baseline EPSP amplitude (baseline mean EPSP amplitude = 0.004 mV, EPSP ratio = 57.7; Fig. 4d). While such small EPSP amplitudes have not been reported for this connection type[67], this could reflect a limitation of paired recordings to detect weak connections given noise, rather than a limitation in the circuit and plasticity models. The model also partially reproduces the experimental results for a pairing protocol at 0.2 Hz[43] (Fig. 4e-g). While for this stimulation protocol we did not observe LTD as reported in vitro, we could not find statistically significant differences between the in vitro and in silico results. Note that again we excluded two outlier connections from the analysis to avoid the skewing effects on the average EPSP ratio from weak connections, as described above (baseline mean EPSP amplitude <0.05 mV; Fig. 4g). Furthermore,

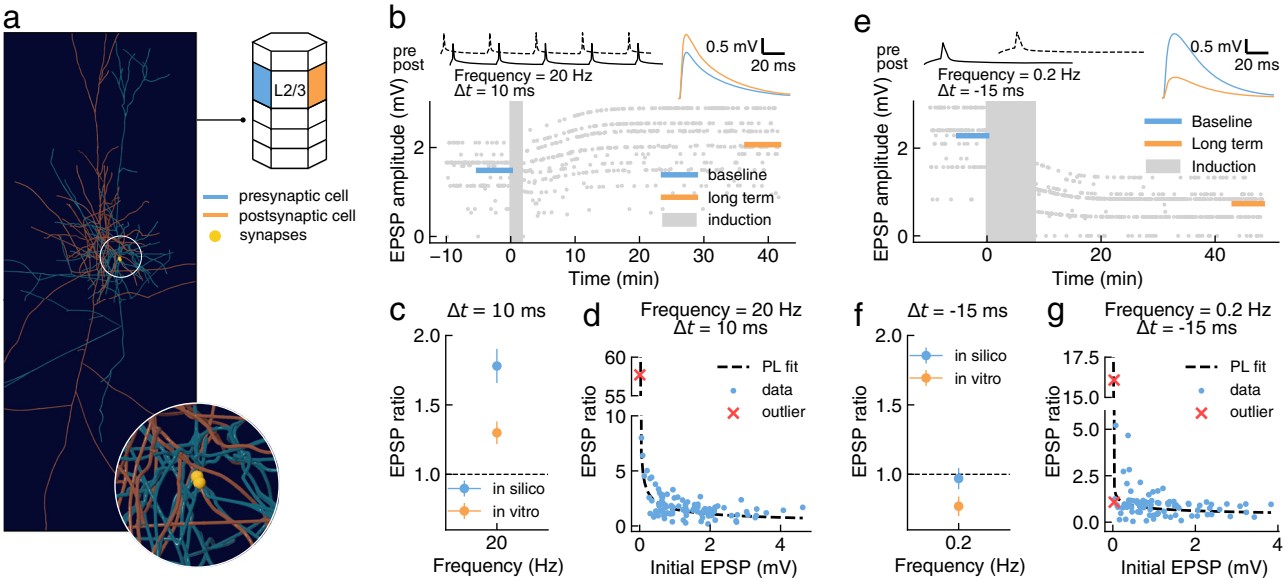

**Fig. 4 Testing plasticity model generalization on the L2/3-PC to L2/3-PC connection type. a** 3-D rendering of a representative pair of connected layer 2/3 pyramidal cell (L2/3-PC) to L2/3-PC in the in silico model. Inset shows a magnified view of the synapses mediating the connection (yellow spheres). **b** Evolution over time of simulated EPSP amplitude (bottom) during a typical plasticity induction protocol (top left; one burst shown out of 10). Mean EPSP amplitudes (top right) are shown before (baseline; blue) and after (long term; orange) the induction protocol. **c** Comparison of in silico and in vitro synaptic changes for pairings at 20 Hz and $\Delta t = 10$ ms ($n = 99$). Welch's unequal variances two-sided t-test showed a significant difference between in vitro and in silico data ($p\text{-value} = 0.002$). Experimental data (in vitro) from Egger et al.[41]. **d** Change of EPSP ratio after plasticity induction as a function of initial EPSP amplitude. Data fit by power law (dashed black line; $f(x) = 1.52x^{-0.49}$). **e** As in **b** for a different stimulation protocol (top left; one pairing shown out of 100). **f** Comparison of in silico and in vitro synaptic changes for pairings at 0.2 Hz and $\Delta t = -15$ ms ($n = 98$). Welch's unequal variances two-sided t-test showed no significant differences between in vitro and in silico data ($p\text{-value} = 0.069$). **g** Change of EPSP ratio after plasticity induction as a function of initial EPSP amplitude. Data fit by power law (dashed black line; $f(x) = 0.74x^{-0.26}$). Experimental data (in vitro, panels **c**, **d**) from Egger et al.[41] and (in vitro panels **f**–**g**) from Banerjee et al.[43]. Population data reported as mean ± SEM. For the full distribution of in silico experiment outcomes see Supplementary Fig. A.12 (data in panels **c**, **d**) and Supplementary Fig. A.13 (data in panels **f**, **g**).

the clear negative correlation between initial connection strength and LTP (Fig. 4g) suggests that our model could be biased towards LTP compared to in vitro reports, as in silico experiments are not limited by noise in the recordings and so weak connections are reliably detected. It should also be considered that this last experiment was performed in mice, while our circuit model[5] and training set is based on rat data. Finally, we tested our model against an in vitro dataset from visual cortex[44], featuring a set of stimulation protocols comparable with those considered so far for L2/3-PC to L2/3-PC connections (Supplementary Fig. A.6). Our model does not reproduce these results, as it showed a greater tendency towards LTP (Supplementary Fig. A.6a, b). However, the plasticity outcomes in this dataset are not entirely consistent with the experimental reports for this connection type in the somatosensory cortex. In particular, the 20 Hz stimulation protocol failed to induce LTP in the visual cortex[44], as was observed in the somatosensory cortex[41]. While the most obvious explanation for this discrepancy would be that different brain areas behave differently in terms of plasticity, we noticed that Zilberter et al.[44] remarked how connections with low release probability were discarded from their experiments. We repeated the analysis of EPSP ratio in our in silico experiments using progressively larger release probability cut offs (i.e. connections whose mean release probability is below a threshold were excluded; Supplementary Fig. A.6c). We found that for a sufficiently large cut off our model could reproduce the experimental results in Zilberter et al.[44]. Although the needed exclusion threshold seems too large to be compatible with any reasonable definition of "weak connections", this result shows how apparently minor differences in the experimental procedures could greatly influence the final results.

Next, we considered in vitro experiments on STDP between L4-PC to L2/3-PC connections[42] (see Supplementary Fig. A.25 for synaptic parameters distributions). Again, we sampled 100 PC connections and reproduced in silico the in vitro protocols[42] (Fig. 5a, b). We found that our model correctly reproduces the experimental results at $\Delta t = -10$ ms (Fig. 5c, left). The in vitro dataset for this connection type also includes pharmacological manipulations testing the hypothesis that LTD at this connection type requires presynaptic NMDARs. Specifically, the NMDAR blocker MK-801 was applied presynaptically before plasticity induction, and found to abolish LTD[42]. We simulated this manipulation by dropping the LTD-induction rate ($\gamma_d = 0$), which led to a quantitative agreement between the in vitro data and the model (Fig. 5c, right), consistent with the notion that presynaptic NMDARs are required to evoke any form of LTD in this connection type. It should be noted that this result may not generalize to other connection types, as there are known connection-type specific differences in expression of presynaptic NMDARs[68] and their role in LTD. For example, L2/3-PC to L2/3-PC connections are not mediated by presynaptic NMDARs[68] and their plasticity is not affected by presynaptic MK-801[43].

Lastly, we considered a set of experiments assessing plasticity between pairs of L4-SSCs in vitro[41] (see Supplementary Fig. A.27 for synaptic parameter distributions). Pairing-induced plasticity at this connection type has been shown to be non-NMDAR dependent, in contrast to pyramidal connections. Consistent with this observation, our model does not reproduce the reported data (Fig. A.7), highlighting its specificity for describing NMDAR dependent forms of plasticity between pyramidal cells.

Taken together, these results suggest that a single plasticity model with shared parameters can capture the plasticity rules (i.e.

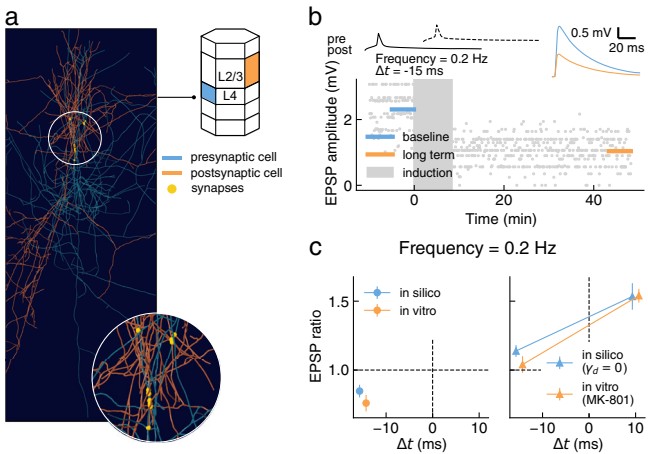

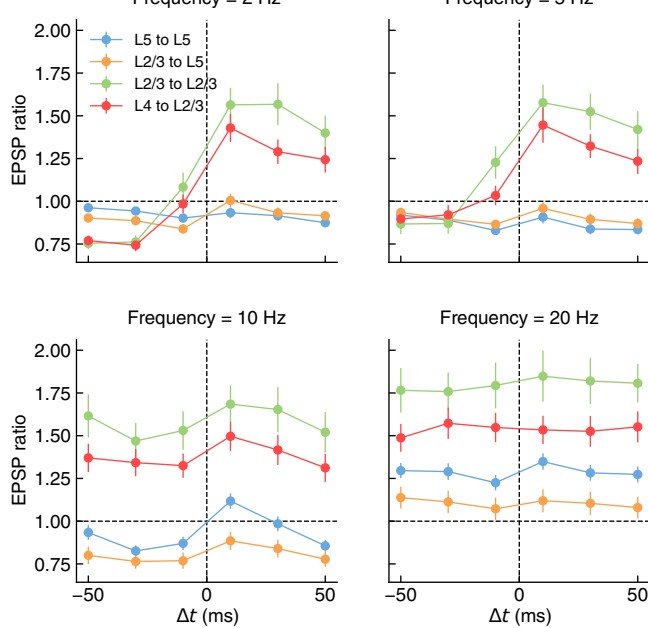

**Fig. 5 Testing plasticity model generalization on the L4-PC to L2/3-PC connection type. a** 3-D rendering of a representative pair of connected L4-PC to L2/3-PC in the in silico model. Inset shows a magnified view of the synapses mediating the connection (yellow spheres). **b** Evolution over time of simulated EPSP amplitude during a typical plasticity induction protocol (top left; one pairing shown out of 100). Mean EPSP amplitudes (top right) are shown before (baseline; blue) and after (long term; orange) the induction protocol. **c** Comparison of EPSP ratios in silico and in vitro for positive and negative timings and with presynaptic NMDAR blocker MK801. Experimental data and simulations without MK801 on the left panel, with MK801 (in vitro) and $\gamma_d = 0$ (in silico) on the right panel. Welch's unequal variances two-sided t-test was n.s. for every protocol (*p-value* from negative to positive stimulation timing: 0.268, 0.209 MK801, 0.959 MK801; $n = 100$). Experimental data (in vitro) from Rodríguez-Moreno and Paulsen[42]. Population data reported as mean ± SEM. For the full distribution of in silico experiment outcomes see Supplementary Fig. A.14 and Supplementary Fig. A.15.

**Fig. 6 Diversity of in silico STDP across stimulation frequencies and connection-types.** Plasticity at connections between L5-TTPC to L5-TTPC (blue, $n = 100$), L2/3-PC to L5-TTPC (orange, $n = 99$, due to an excluded outlier), L2/3-PC to L2/3-PC (green, $n = 100$) and L4-PC to L2/3-PC (red, $n = 100$) was induced following stimulation protocols in Markram et al.[40] to allow for a standardized comparison of STDP curves. Four different pairing frequencies were considered: 2 Hz (top left), 5 Hz (top right), 10 Hz (bottom left) and 20 Hz (bottom right). Data for L5-TTPC to L5-TTPC as in Fig 2. All panels report EPSP ratio data as mean ± SEM. For the full distribution of in silico experiment outcomes see SI A.10 (L5-TTPC to L5-TTPC), A.16 (L2/3-PC to L5-TTPC), A.17 (L2/3-PC to L2/3-PC), A.18 (L4-PC to L2/3-PC).

outcomes of protocols) of novel PC connection types and manipulations never seen during optimization, and predict plasticity outcomes for all other connection types in the juvenile rodent sensory cortex.

**Connection-type specific STDP**. The diversity of plasticity outcomes in the neocortex could indicate the existence of connection-type specific learning rules, which together form a coordinated learning algorithm. Alternatively, and more trivially, it could merely reflect the heterogeneity of in vitro protocols and preparations reported in the literature, and simulated here. While differential plasticity rules have been reported between at least two PC connection types[36], whether this is a general cortical principle remains unclear when comparing experimental results across the diverse PC connection types[69].

To investigate this, we generated a standardized in silico map of STDP for the PC connection types considered in this work. Our results show the existence of three qualitatively different STDP curves for these connection types (Fig. 6). L5-TTPC to L5-TTPC exhibited strong sensitivity to spike-timing at intermediate stimulation frequencies (10 Hz), as expected from in vitro results[12] and already discussed in the previous section (see Fig. 3). Plasticity of L4-PC to L2/3-PC and L2/3-PC to L2/3-PC connections was instead maximally affected by spike-timing at low stimulation frequencies (2 and 5 Hz). Finally, L2/3-PC to L5-TTPC connections did not show any STDP window for the stimulation protocols herein considered.

To dissect the origin of the observed connection-type specificity, we analyzed the relationship between synaptic parameters and plasticity outcomes for the 10 Hz, 10 ms STDP

protocol. As all PC types in this work share the same set of parameters for the plasticity model, we hypothesized that these different STDP outcomes could be explained by the specificity of the innervation and synaptic transmission dynamics of each connection type, as prescribed by the circuit model[5]. In particular, the four connection types considered here differ primarily in their apical ratio (the fraction of synapses on apical dendrites over the total number of synapses in the connection), and their NMDAR conductance (see Fig. A.8). Furthermore, we found that apical ratio and NMDAR conductance are strongly correlated with EPSP ratio and so are good candidates to explain the observed connection-type specificity (Fig. A.8; bottom center, bottom right).

Consistent with our hypothesis, we found that each of the three unique STDP curves could be associated to different clusters of these two parameters (see Fig. A.9). Connections between L4-PC to L2/3-PCs and L2/3-PC to L2/3-PCs exhibit similar apical ratios and NMDAR conductances, and have comparable plasticity outcomes. Compared to these two connection types, connections between L5-TTPCs have larger conductances. To explain how larger conductances result in smaller levels of LTP, we must consider the strong positive correlation between conductance, spine volume and RRP size and how plasticity thresholds are calibrated in this model. As previously explained, these thresholds are computed as linear combinations of the calcium peak during a single EPSP assuming all vesicles are released, $C_{pre}$, and the calcium peak during a single bAP, $C_{post}$. Due to stochastic vesicle release, the probability of obtaining a calcium transient as large as $C_{pre}$ is proportional to

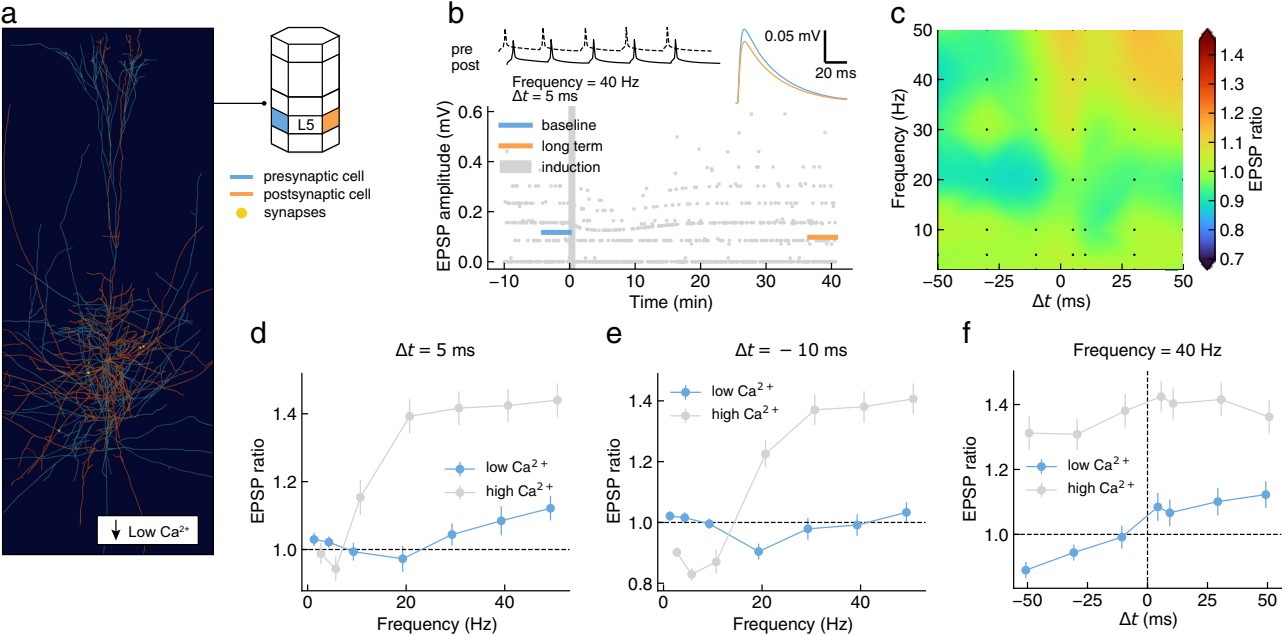

**Fig. 7 Prediction of plasticity outcomes under physiological extracellular calcium conditions (in vivo-like; low [Ca²⁺]ₒ). a** 3-D rendering of a representative pair of connected L5-TTPC in the in silico model. **b** Evolution over time of simulated EPSP amplitude (bottom) during a typical plasticity induction protocol (top left; one burst shown out of 10). Mean EPSP amplitudes (top right) are shown before (baseline; blue) and after (long term; orange) the induction protocol. **c** Timing and frequency dependence of LTP/LTD in silico for the protocol described in **b** ($n = 100$). Simulated configurations are shown as black markers, with cubic interpolation elsewhere. **d** Comparison of frequency dependence of synaptic changes for high and low calcium at $\Delta t = +5$ ms. Welch's unequal variances two-sided t-test showed significant differences for six out of seven protocols (p-value from low to high stimulation frequency: 0.260, 0.045, 0.007, < 0.001, < 0.001, < 0.001, < 0.001; $n = 100$). **e** Comparison of frequency dependence of synaptic changes for high and low calcium at $\Delta t = -10$ ms. Welch's unequal variances two-sided t-test showed significant differences for every protocols (p-value from low to high stimulation frequency: < 0.001, < 0.001, 0.004, < 0.001, < 0.001, < 0.001, < 0.001; $n = 100$). **f** Comparison of STDP for high and low calcium at a frequency of 40 Hz. Welch's unequal variances two-sided t-test showed significant differences for every protocols (all p-values < 0.001; $n = 100$). High calcium data as in Fig. 3a-f. Population data reported as mean ± SEM. For the full distribution of in silico experiment outcomes see SI A.10 and A.19.

$U_{SE}^{N_{RRP}}$ (i.e. $p^n$ for a binomial model). Since $0 < U_{SE} < 1$ by definition, it follows that larger synapses are less likely to reach plasticity thresholds during normal activity than smaller ones.

Compared to the other three connection types, connections between L2/3-PCs and L5-TTPCs, are mostly made by apical synapses. Experimentally, these are more prone to depression than basal synapses for the plasticity induction protocols considered here. In our model, apical and basal synapses were parameterized independently to reproduce this result. As a consequence, such connections that are composed mostly of apical synapses are biased towards depression. Note that spine volume and RRP size in this model are strongly correlated with the synaptic conductance, and so we only considered the latter for this analysis. The existence of these distinct STDP classes could provide the neocortex with a set of complementary plasticity rules and support the implementation of complex learning strategies.

**Plasticity at physiological calcium conditions**. Extracellular calcium concentration is an important modulator of synaptic transmission and calcium currents. Calcium levels in vivo are significantly lower than the conditions of plasticity experiments in vitro considered in this work to constrain and test the model (in vivo: 1 to 1.3 mM; in vitro: 2 mM or higher[70]). Given the central role of calcium for plasticity, it is important to take into account the impact of physiological calcium concentration to understand the learning rules which are operating in vivo.

However, plasticity remains largely unexplored under such experimental conditions. To our knowledge no studies have been reported on this issue in neocortex, and only one exists for the hippocampus[39].

To quantify the impact of a physiological calcium concentration on plasticity induction protocols, we repeated the L5-TTPC to L5-TTPC simulations correcting release probabilities, thresholds, and calcium currents to 1.2 mM calcium (Fig. 7a, b, see Methods for in vivo correction details). Under these conditions, we found that LTP/LTD magnitudes were significantly reduced (Fig. 7c) and plasticity required higher frequency stimulation to be induced (30 Hz and above for LTP, Fig. 7d; 20 Hz for LTD, Fig. 7e). Furthermore, at anti-causal temporal relationships ($\Delta t = -10$ ms), we found that potentiation at high frequencies was abolished (Fig. 7e). Surprisingly, we found that bidirectional STDP appears at 40 Hz and above, in contrast to high calcium conditions where only LTP was observed at high frequency stimulation (Fig. 7f).

Taken together, these predictions indicate that differences in calcium concentration between in vitro and in vivo conditions could have a dramatic effect on plasticity outcomes, yielding a qualitatively different perspective on plasticity rules at play in vivo.

## Discussion

In this study we designed a calcium-based model of synaptic plasticity capable of reproducing the results of LTP/LTD

experiments on L5-TTPCs, and showed that this model could be transplanted as-is to predict synaptic plasticity data for other PC connections in the somatosensory cortex. Our formalism accounts for the physiological heterogeneity of synaptic variables, such as release probability and conductance, and provides a quantitative description of the behavior of individual synapses during plasticity of whole (multi-synaptic) connections. We used our model to analyze the relationship between spike-timing, frequency and synaptic plasticity at in vivo levels of calcium, predicting important differences compared to typical in vitro conditions. Our results suggest that postsynaptic calcium diversity is sufficient to explain connection-type specificity of activity-dependent synaptic plasticity between PCs in the neocortex. Our proposed model and fitting procedure offer a method to estimate plasticity outcomes for connections that have yet to be experimentally explored, providing a candidate unifying theoretical framework for studying cortical learning algorithms in silico.

Our predictions of plasticity at in vivo levels of calcium highlighted major qualitative differences with respect to in vitro conditions, as also suggested by a recent experimental study in the hippocampus[39]. While we accounted for the effects of reducing extracellular calcium on multiple components of the synapse model (i.e. calcium driving force, plasticity thresholds and NMDAR fractional calcium current), these results are mostly due to the estimated five-fold decrease in synaptic release probability[70]. Under these conditions, successful pairing events become rare, suggesting an important role for N-methyl-D-aspartate (NMDA) spikes and other dendritic nonlinearities for evoking sufficiently large calcium influxes to induce plasticity[71–73]. The proposed in silico framework could be used to study these dynamics without major modifications, for example by simulating the scenario where multiple presynaptic neurons activate neighboring synapses. In this scenario, synapses could cooperate through voltage nonlinearities to evoke calcium transients of sufficient magnitude to induce learning in a single trial (one-shot learning[74,75]), circumventing the high failure rate.

Many other models of synaptic plasticity are available in the literature and they could be in principle compatible with the circuit model and the training data used in this work (for a review see Manninen et al.[25]). Here we focused on finding a solution to the problem of parameterizing the large set of heterogeneous synapses across the different PC connection types in neocortex, given the sparse experimental constraints available, and with the particular aim of generalizing to physiological levels of calcium. The commonly used STDP models, although appealing for their simplicity, are not an appropriate framework for our aim as they would require individualized fitting for each connection type, for which experimental data are lacking. For this reason, we decided to adapt a previously published calcium-based model[19] to match the description level of our neural tissue model[5]. This approach allowed us to take into account important aspects of synaptic physiology often neglected in plasticity studies, such as stochastic vesicle release, dendritic integration, and multi-synaptic connections, while limiting the free parameters to constrain to those concerning the plasticity model. Furthermore, a direct dependence on calcium dynamics allowed the extrapolation of the plasticity model to physiological calcium concentrations without refitting, based on biophysical considerations alone. Several ingredients of the Graupner and Brunel[19] model are based on well-established ideas in the field. For example, the model takes advantage of the full calcium time course, rather than just the peaks, to identify plasticity-inducing events, as in Rubin et al.[17]. Here, we introduced a calcium integrator to associate synaptic events beyond the duration of individual calcium transients, and to compensate for the high synaptic failure rates. The original model proposed by Graupner and Brunel[19] lacks this term as the same effect could be obtained by reliable vesicle release and/or a longer calcium time constant. A similar approach was taken in other studies[17,21,76]. Compared to these other models, our approach highlights the importance of considering the stochasticity of synaptic release, and its calcium dependence, to generalize plasticity rules across different connection types and experimental conditions.

However, our study has several limitations. We only consider NMDAR-dependent forms of LTP/LTD at pyramidal cell connections. Although this family includes the vast majority of connections in the neocortex, other connection types (i.e. inhibitory plasticity) are still likely to play a major role in learning and memory processes. While it may be possible to generalize the approach presented here to these other connections types, it was beyond the scope of the present study. Our model describes the early stages of LTP/LTD, but does not cover the mechanisms of synaptic tagging and consolidation[77], when protein synthesis and competition for molecular resources could potentially override the initial outcome of plasticity induction[78] (for a review see Redondo and Morris[79]). Accounting for these mechanisms would require an extension of the model to include consolidation dynamics on longer timescales (see e.g. Clopath et al.[80]). However, parameterizing such a model in a connection type-specific manner would also presuppose the availability of connection type-specific experimental data in the neocortex, which is currently lacking. Furthermore, we assumed a single synaptic efficacy variable $\rho$, driving both pre- and post-synaptic changes at individual synapses. While under this simplifying assumption the model could reproduce in vitro results on the locus of expression of LTP for whole connections, it could not reproduce presynaptic-only LTD[32]. This could indicate either a transient or long-term violation of the single efficacy assumption. The former case would require a minor extension of our model to be captured, whereas the latter would require modeling separate pre- and post-synaptic efficacies, as explored in a recent theoretical study[81]. It is worth noting that simply masking postsynaptic LTD is not a viable solution, as postsynaptic LTD has been reported for other induction paradigms[82–85]. Our model also assumes that the number of release sites at each synapse is fixed, and unaffected by plasticity. This is consistent with the classical view of presynaptic plasticity being primarily mediated by changes of release probability[63,86]. However, a recent report on the nanoscale organization of synapses has revealed the existence of synaptic nanomodules, that is, a tight coupling of the pre- and post-synaptic machinery responsible for synaptic transmission (i.e. the presynaptic active zone (AZ) and postsynaptic receptors)[87]. These pre- and post-synaptic components have been shown to be added in pairs as modules during LTP, and so proposed as atomic building blocks of plasticity at the synaptic level[87]. Extending our model to account for these dynamics would require more detailed experimental data on the relative contributions of AZ-plasticity and release probability changes to LTP/LTD.

A key assumption of this work is the bistability of pre- and postsynaptic plasticity. Studies at individual synapses in the hippocampus show that postsynaptic (AMPAR-mediated) plasticity is bistable and synapses jump between extreme states of efficacy[88,89]. Plasticity of presynaptic changes, on the other hand, has been shown in the hippocampus to support multiple stable states, or evolve on a continuum[90]. While we could have opted for a different model to describe presynaptic stability of plastic changes, we judged the available experimental evidence insufficient to justify a more complex model. Accounting for this difference could yield "overfitting" and would be inconsistent with our previous simplifying assumption that pre- and postsynaptic changes are matched on long time scales. Furthermore, this bistability assumption, in contrast to a model based on a

continuum of stable states, provides some protection against sporadic plasticity threshold crossings, and thereby a mechanism for long-term stability of learned configurations despite ongoing spontaneous activity in networks. That is, in our model long-term plasticity is induced either by accumulating changes over a short period of time or in one-shot via extremely large calcium events, such as a calcium spike. On the other hand, sparse and weak threshold crossing events are quickly forgotten as they are not sufficient to drive the synaptic efficacy $\rho$ out of its current basin of attraction, which is consistent with in vitro observations during mild stimulation (see Fig. 3d, 2 and 5 Hz stimulation protocols).

We parameterized and tested our model using experimental results reported in the literature. As our objective was to investigate type-specific plasticity using an in silico model of rat somatosensory cortex[5], we targeted experimental data from juvenile rat S1 where pre- and post-synaptic neuron types were identified. However, due to the sparsity of experimental sources, we also included data from rat visual cortex for training purposes, and from mouse visual and somatosensory cortices for testing. Due to the quantitative nature of the present modeling study, several prominent data sets from adult animals or non-neocortical regions have been excluded. For example, Letzkus et al.[38] also reported a dependence of plasticity on dendritic location which is qualitatively consistent with the dataset we compared to here[36], but we didn't pursue a quantitative comparison to this dataset because it was from older animals, and plasticity induction is known to be significantly age dependent[91].

We do not claim that the available experimental data is sufficient to completely constrain the model or validate its predictive power. For example, apical synapses are mostly depressing in our model under the protocols considered and the L2/3-PC to L5-TTPC connection type, dominated by apical synapses, lacked a clear STDP window. This could be a specific feature of apical synapses, but more data would be required to support such a conclusion. That is, we had experimental data for only one connection type and protocol featuring mostly apical synapses, and it was not from somatosensory cortex but rather the visual cortex[36], which could be more prone to LTD (cf. Egger et al.[41] and Zilberter et al.[44]). Moreover, the data available for testing model predictions is limited in terms of connection types and protocols. However, the goal of this study was to provide a candidate null model until more comprehensive experimental characterization becomes available. We argue that such an integrative approach maximizes the value of the few experimental data points available, thereby helping to homogenize the results, detect anomalies and extrapolate beyond them. Further experiments would be valuable to test predictions of the model, and refine its assumptions.

The extensive testing on data from the visual cortex highlighted several quantitative differences between the model predictions and in vitro experiments, possibly suggesting a strong regional dependence of the plasticity model parameterization. While we cannot rule out this possibility, we could also imagine a scenario where synaptic physiology, cell morphology and innervation profiles could quantitatively account for the observed inter-regional differences, without the need for region-specific parameterization of the plasticity model. That is, as PC type-specific plasticity in the somatosensory cortex could be explained by our model using a non-type-specific set of parameters, similarly region-type-specific plasticity could emerge from regional differences other than plasticity itself. Considering the many anatomical similarities between PCs in the different sensory cortices, we hypothesize that adapting the synaptic transmission properties of our in silico connections (e.g. the release probabilities, conductances, depression and facilitation time constants) to region-specific in vitro data could be sufficient to reconcile the observed

differences in plasticity outcomes. While testing this hypothesis is beyond the scope of this work, given the sparsity of data, our study still provides an initial candidate framework for generalizing plasticity principles across cortical regions.

Optimizing the plasticity model is a computationally expensive procedure, exceeding the capabilities of a typical workstation. However, re-optimization should not be required for most researchers wishing to make use of the plasticity model in their own studies. We provide a set of parameters optimized for the somatosensory cortex and, since generalization without re-fitting is one of the main results of this work, it is possible that the model and parameters provided here could be reused to describe plasticity in other cortical areas and species. For cases where researchers would want to re-optimize the model on their own datasets (i.e. different brain regions, new plasticity experiments) or undertake an extension of the model, we provide the plasticity optimization source code. Furthermore, a few modifications and approximations of our methods could potentially reduce the computational cost. For example, the initial mean EPSP amplitude for each in silico connection could be pre-computed and cached, allowing the initial part of each simulation to be skipped. Similarly, one might fit a function to approximate the relationship between synaptic parameters and mean EPSP amplitude, and simulate only the plasticity induction part of the experimental protocol. Lastly, it could be possible to obtain a significant speedup by running the optimization on a GPU using CoreNEURON[92], a recent development of the NEURON simulator.

The present work highlights the value of considering dendrites and calcium dynamics explicitly when studying synaptic plasticity. Using digital reconstructions of pyramidal neurons was key to generalize the results of a few experiments to produce a dense predictive map of plasticity between PCs in the somatosensory cortex. Moreover, directly modeling the physiology of postsynaptic calcium transients and their role in synaptic plasticity provided a means to extrapolate the outcome of traditional LTP/LTD induction protocols to more realistic in vivo conditions from biophysical considerations. In particular, we showed that reducing calcium from in vitro to physiological levels profoundly alters LTP/LTD dynamics, predicting a vast unexplored experimental territory of plasticity in vivo. Furthermore, while STDP-like pairing protocols are a very effective tool to investigate the landscape of synaptic changes, they are restricted to pre-post timing relationships. It is becoming increasingly appreciated that NMDA spikes and other dendritic non-linearities triggered by more complex spiking activity motifs play an important role in learning and perception in vivo[71,72,85,93–99]. By providing a quantitative approach to determine plasticity outcomes from calcium dynamics in a model of cortical tissue, we offer a way to predict learning rules between PCs which have yet to be characterized experimentally. The result is an integrated modeling framework well suited to explore the input/output conditions relevant for plasticity in vivo and ultimately the biological algorithms of learning in the neocortex.

## Methods

The long-term potentiation (LTP)/long-term depression (LTD) model used in this work extends the alpha-amino-3-hydroxy-5-methyl-4-isoxazole propionate receptor (AMPAR) / N-methyl-D-aspartate receptor (NMDAR) synapse described in Markram et al.[5]. For the sake of completeness, we present in the following sections all the components of the synapse model, including those previously developed[5]. Unless otherwise indicated in the text, model parameters for each connection type are those due to a 2018 internal release of the model available online at: https://bbp.epfl.ch/nmc-portal/microcircuit. Part of the text in the following sections is adapted from the doctoral dissertation of the first author Chindemi[100].

**AMPAR**. The AMPARs are described as a point current source using a double exponential conductance profile[5,45,46]:

$$I_{AMPAR}(t) = G_{AMPAR}(t) \cdot \left( V(t) - E_{AMPAR} \right) \tag{6}$$

$$G_{AMPAR}(t) = \hat{G}_{AMPAR}(t) \cdot (B(t) - A(t)) \tag{7}$$

$$\frac{d}{dt} A = -\frac{1}{\tau_r} A + \varphi \cdot \frac{N_{rel}}{N_{sites}} \cdot \delta(t - t_{rel}) \tag{8}$$

$$\frac{d}{dt} B = -\frac{1}{\tau_d} B + \varphi \cdot \frac{N_{rel}}{N_{sites}} \cdot \delta(t - t_{rel}) \tag{9}$$

$$\varphi = -e^{-t_{peak}/\tau_r} + e^{-t_{peak}/\tau_d} \tag{10}$$

$$t_{peak} = \frac{\tau_r \tau_d}{(\tau_d - \tau_r) \log(\tau_d/\tau_r)}, \tag{11}$$

where $I_{AMPAR}$ is the current produced by the synaptic population of AMPARs; $E_{AMPAR}$ is the reversal potential of the receptor; $V$ is the membrane potential; $G_{AMPAR}$ is the conductance of the receptor population, with peak $\hat{G}_{AMPAR}$ which evolves by plasticity as described below; $A$ models the rising component of the conductance, with time constant $\tau_r$; $B$ models the decaying component of the conductance, with time constant $\tau_d$; $t_{rel}$ is the time of a successful release event; $N_{rel}$ is the number of vesicles released at time $t_{rel}$ and $N_{sites}$ is the total number of release sites (see section Neurotransmission); $t_{peak}$ is the time to peak of the conductance and $\varphi$ is a normalization factor such that $G_{AMPAR}(t_{rel} + t_{peak}) = \hat{G}_{AMPAR}$. All the parameters of the AMPAR model are prescribed in the neocortical tissue model[5], with the exception of the peak AMPAR conductance $\hat{G}_{AMPAR}$, which is a dynamic variable changing with synaptic plasticity as described below in section Coupling long-term plasticity and synaptic transmission. Please notice that we dropped the AMPAR subscript for $A$, $B$, $\varphi$, $t_{peak}$, $\tau_r$ and $\tau_d$ to improve readability, but these variables and parameters are independent from those of the NMDAR, described in the next section.

**NMDAR**. The NMDARs are described as a point current source using a double exponential conductance profile and incorporating the voltage dependence due to the magnesium block[5,45,46,50]:

$$I_{NMDAR}(t) = m \cdot G_{NMDAR}(t) \cdot \left( V(t) - E_{NMDAR} \right) \tag{12}$$

$$m = \frac{1}{1 + ([Mg^{2+}]/\theta)e^{-\kappa V}} \tag{13}$$

$$G_{NMDAR}(t) = \hat{G}_{NMDAR} \cdot (B(t) - A(t)) \tag{14}$$

$$\frac{d}{dt} A = -\frac{1}{\tau_r} A + \varphi \cdot \frac{N_{rel}}{N_{sites}} \cdot \delta(t - t_{rel}) \tag{15}$$

$$\frac{d}{dt} B = -\frac{1}{\tau_d} B + \varphi \cdot \frac{N_{rel}}{N_{sites}} \cdot \delta(t - t_{rel}) \tag{16}$$

$$\varphi = -e^{-t_{peak}/\tau_r} + e^{-t_{peak}/\tau_d} \tag{17}$$

$$t_{peak} = \frac{\tau_r \tau_d}{(\tau_d - \tau_r) \log(\tau_d/\tau_r)}, \tag{18}$$

where $I_{NMDAR}$ is the current produced by the synaptic population of NMDARs; $E_{NMDAR}$ is the reversal potential of the receptor; $V$ is the membrane potential; $m$ is the magnesium block gating variable[50]; $\theta$ is an appropriate scaling factor of the extracellular magnesium concentration $[Mg^{2+}]$; $\kappa$ is the slope of magnesium voltage dependence; $G_{NMDAR}$ is the conductance of the receptor population, with peak $\hat{G}_{NMDAR}$; $A$ models the rising component of the conductance, with time constant $\tau_r$; $B$ models the decaying component of the conductance, with time constant $\tau_d$; $t_{rel}$ is the time of a successful release event; $N_{rel}$ is the number of vesicles released at time $t_{rel}$ and $N_{sites}$ is the total number of release sites (see section Neurotransmission); $t_{peak}$ is the time to peak of the conductance and $\varphi$ is a normalization factor such that $G_{NMDAR}(t_{rel} + t_{peak}) = \hat{G}_{NMDAR}$. Note that in this work we do not account for plasticity of NMDARs or for homeostatic maintenance of the AMPAR to NMDAR ratio. Please notice that we dropped the NMDAR subscript for $A$, $B$, $\varphi$, $t_{peak}$, $\tau_r$ and $\tau_d$ to improve readability, but these variables and parameters are independent from those of the AMPAR, described in the section above.

Equation (13) describes the dynamics of magnesium gating using the formalism proposed in Jahr and Stevens[50]. As the original values of the slope parameter $\theta$ and the scale parameter $\kappa$ reported in Jahr and Stevens[50] were fit to hippocampal data, we re-estimated these parameters using recent data for the neocortex[51]. In particular, we computed $\theta = 0.072$ and $\kappa = 2.552$ by inspection from the steady state fraction of unblocked conductance in Vargas-Caballero and Robinson[51],

described there in the Boltzmann formalism as

$$b_\infty(V) = \frac{1}{1 + \exp\left( -(V - V_{0.5}) z \delta F / RT \right)}, \tag{19}$$

where $V$ is the membrane voltage; $V_{0.5} = -13\,mV \pm 2.45$ is the voltage for half-maximal block; $z = 2$ is the valence of magnesium ions; $\delta = 0.96 \pm 0.01$ is the fractional sensitivity of the block to membrane voltage; $T = 36\,°Celsius$ is the temperature; $F$ and $R$ have their usual thermodynamical meaning. Experiments were carried on at 1 mM magnesium concentration and voltage traces were corrected for liquid junction potential[51]. It is interesting to notice that fitting $m$ to data recorded from neocortical neurons and correcting for liquid junction potential[51] actually produces a curve very similar to the one derived from theoretical considerations by Rhodes[101] to facilitate the generation of N-methyl-D-aspartate (NMDA)-spikes, which are known to be difficult to induce using the Jahr and Stevens[50] model (see Fig. A.29a).

To estimate calcium concentration at the synapse, we needed to determine NMDAR-mediated calcium currents. In Schneggenburger et al.[102], the fractional calcium current through the NMDAR, $P_f$, was calculated from the Goldman–Hodgkin–Katz (GHK) flux equation as follows

$$P_f = \frac{I_{Ca}}{I_{Ca} + I_M} = \frac{4[Ca^{2+}]_o}{4[Ca^{2+}]_o + \frac{p_M}{p_{Ca}}[M](1 - e^{2\frac{VF}{RT}})}, \tag{20}$$

where $I_{Ca}$ and $I_M$ are the currents due to calcium ions and due to all monovalent ions, respectively; $[Ca^{2+}]_o$ is the extracellular calcium concentration; $[M]$ is the concentration of monovalent ions, assumed to be identical inside and outside the cell membrane; $p_{Ca}$ and $p_M$ are the permeabilities to calcium and monovalent ions, respectively; $V$ is the membrane voltage; $T = 36.85\,°Celsius$ is the temperature; $F$ is the Faraday constant and $R$ is the ideal gas constant (all parameter values as in Schneggenburger et al.[102]).

However, equation (20) cannot be used directly to determine the calcium current from the total NMDAR-mediated current (i.e. by multiplying $P_f$ and $I_{NMDAR}$), since $I_{NMDAR} = 0$ around the reversal potential $E_{NMDAR}$, and so the resulting calcium current would be erroneously estimated to be zero for certain voltages in our range of interest. To circumvent this issue, we modeled the calcium influx through NMDARs as a separate current mediated by an extracellular-concentration-dependent fraction of the total conductance, resulting in a current of the form

$$\tilde{I}_{NMDAR}(t) = \tilde{G}_{NMDAR}(t) \cdot \left( V(t) - \tilde{E}_{NMDAR} \right) \tag{21}$$

$$\tilde{G}_{NMDAR}(t) = s([Ca^{2+}]_o) \cdot G_{NMDAR}(t), \tag{22}$$

where $V$ is the membrane potential; $\tilde{E}_{NMDAR}$ is the reversal potential of the current, fixed here to 40 mV based on experimental observations in Schneggenburger et al.[102] and assumed for simplicity to be independent of extracellular calcium in the range $1 - 2\,mM$; $G_{NMDAR}$ is the total NMDAR conductance given by equation (14) and $s([Ca^{2+}]_o)$ represents the calcium fraction of the total conductance as a function of extracellular calcium concentration. As we assumed $s([Ca^{2+}]_o)$ is independent of membrane voltage, we modeled it as proportional to $P_f$ evaluated at large negative membrane potentials as follows

$$s([Ca^{2+}]_o) = \alpha \cdot \lim_{V \to -\infty} P_f([Ca^{2+}]_o), \tag{23}$$

where $\lim_{V \to -\infty} P_f([Ca^{2+}]_o)$ is the fractional calcium current at very negative potentials, and $\alpha = 0.6$ is a scaling factor which was tuned to provide approximate self-consistency between currents determined by this model and equation (20) (see Fig. A.29).

**Neurotransmission**. The dynamics of synaptic vesicle release is described using a stochastic version of the canonical Tsodyks–Markram (TM) model with multi-vesicular release (MVR)[5,45,46,48,86]. It is effectively analogous to a traditional binomial model of vesicle release $B(N_{RRP}, U(t))$, where $N_{RRP}$ is the number of vesicles available at any given moment in the readily-releasable pool (RRP) (out of $N_{sites}$, the total number of release sites) and $U(t)$ is the dynamical release probability of the TM formalism which exhibits short-term facilitation dynamics described as follows

$$U(t) = U(t_{syn}) \cdot e^{-(t-t_{syn})/\tau_{fac}} + U_{SE} \cdot (1 - U(t_{syn}) \cdot e^{-(t-t_{syn})/\tau_{fac}}), \tag{24}$$

where $U_{SE}$ is the stable fixed point of the release probability in the absence of stimulation; $t_{syn}$ is the time of the last presynaptic spike; $\tau_{fac}$ is the facilitation time constant. Release of a vesicle decrements $N_{RRP}$, and increments $N_R$, which is the number of vesicles awaiting recovery.

Vesicle recovery also follows a binomial model $B(N_R, 1 - P_{surv}(t))$ with the survival probability of the un-recovered state given by

$$P_{surv}(t) = e^{-(t-t_{syn})/\tau_{dep}}, \tag{25}$$

where $\tau_{dep}$ is the depression time constant. The total number of release sites, $N_{sites}$, was constrained extending the methods in Barros-Zulaica et al.[48] to the range of

2-4 vesicles[61]. All the parameters of the synaptic transmission model are prescribed in the neocortical tissue model[5], with the exception of the stable fixed point of the release probability $U_{SE}$, which is a dynamic variable changing with synaptic plasticity as described below in section Coupling long-term plasticity and synaptic transmission. Note that in this work we do not account for plasticity of $N_{sites}$.

**VDCC.** In this work, a simple inactivating population of R-type voltage-dependent calcium channels (VDCCs) in spines was modeled as a point current in the Hodgkin-Huxley (HH) formalism[53,54] as follows

$$I_{VDCC}(t) = G_{VDCC}(t) \cdot (V(t) - E_{VDCC}) \tag{26}$$

$$G_{VDCC}(t) = \hat{G}_{VDCC} \cdot m^2 h \tag{27}$$

$$\hat{G}_{VDCC} = 4\pi \bar{g} \left( \frac{3}{4\pi} X \right)^{2/3} \tag{28}$$

$$\frac{d}{dt} m = \frac{m_\infty(V) - m}{\tau_m} \tag{29}$$

$$\frac{d}{dt} h = \frac{h_\infty(V) - h}{\tau_h} \tag{30}$$

$$m_\infty(V) = \frac{1}{1 + e^{\frac{V_{hm} - V}{k_m}}} \tag{31}$$

$$h_\infty(V) = \frac{1}{1 + e^{\frac{V_{hh} - V}{k_h}}}, \tag{32}$$

where $I_{VDCC}$ is the current produced by the channel population; $V$ is the membrane potential; $E_{VDCC}$ is the reversal potential for calcium; $G_{VDCC}$ is the conductance of the population, with peak $\hat{G}_{VDCC}$, calculated assuming a spherical spine head; $\bar{g} = 0.0744\ nS/\mu m^2$ is the VDCC surface area density[54,103]; $X$ is the spine head volume (see section Dendritic spines); $m$ is the activation variable, with time constant $\tau_m = 1\ ms$ (chosen to approximate the rising phase in Magee and Johnston[53]) and steady state $m_\infty$; $h$ is the inactivation variable, with time constant $\tau_h = 27\ ms$ (Magee and Johnston[53]; corrected to 34 °Celsius) and steady state $h_\infty$; $V_{hm} = -5.9\ mV$ is the half-maximum activation voltage[53] and $k_m = 9.5\ mV$ is the slope factor[53]; $V_{hh} = -39\ mV$ is the half-maximum inactivation voltage[53] and $k_h = -9.2\ mV$ is the slope factor[53].

**Dendritic spines.** The vast majority of excitatory synapses are located on dendritic spines. With respect to synaptic plasticity, spines are particularly important because they act as biochemical compartments, encapsulating and localizing the molecular machinery responsible for implementing synaptic changes. Furthermore, spine volume is a key determinant of intracellular calcium concentration transients arising from calcium currents. In this work we model spines as point processes rather than explicitly modeling spine compartments for computational reasons, but we do account for biochemical compartmentalization and volume-related effects on calcium concentration. Specifically, we assume a spherical spine head, and calcium ions are not allowed to diffuse into the parent dendrite. Estimates for the spine head volume distribution have been obtained from intracellularly injected basal dendrites of 44 layer 5 pyramidal cells of the hindlimb somatosensory cortex of P14 rat, that were 3D reconstructed from high-resolution confocal image stacks. At this age, dendrites show both spines and filopodia, which are long, thin protrusions lacking a bulbous head (Fig. A.28a). Measurements of head volume, head diameter and mean neck diameter were obtained from 8423 reconstructed single dendritic protrusions using the software Imaris (Bitplane AG, Zurich, Switzerland) and were corrected for medium shrinkage as in Toharia et al.[104] and Rojo et al.[105]. Spine measurements are publicly available at http://cajalbbp.es/storage/P14%20excels%20layers%20III%20and%20V. Evaluating the reconstructed data (Fig. A.28a) and the head-to-neck diameter (HND) ratio (Fig. A.28b) revealed that many dendritic protrusions did not exhibit a clear head. To exclude these potential filopodia, only spines that had a head diameter that was larger than the respective neck diameter were used for the analysis (Fig. A.28b, black line). The majority of the excluded dendritic protrusions exhibited a small head volume < 0.05 μm³ (Fig. A.28c, green) while the remaining spines with marked head (Fig. A.28c, blue) showed a head volume of 0.087 ± 0.088 μm³ which could be best approximated by a log-normal distribution ($\mu = -2.8$ and $\sigma = 0.87$). In the neocortical circuit model, we prescribed the spine volume distribution for each connection type as a function of the prescribed synaptic conductance ($\hat{G}_{AMPAR} + \hat{G}_{NMDAR}$) distribution so as to be consistent with the experimentally observed head volume distribution. Specifically, we found that the experimentally observed spine head volume distribution and the prescribed synaptic conductance distribution for spines on basal dendrites of layer 5 thick-tufted pyramidal cells (L5-TTPCs) were related by a linear transformation determined by the ratio of the means of the two distributions. We then generalized this relationship to all connection types by assuming the same linear transformation determined for spines on layer 5 basal dendrites. Experimentally observed correlations between these two synaptic parameters were introduced at

sampling time as described in the section Correlation of synaptic parameters. Furthermore, we determined VDCC conductances on the spine by assuming a spherical spine head and a uniform density of channels on the membrane according to previous estimates[54]. This approach has the consequence that the VDCC conductance is proportional to that of NMDAR for every synapse.

**Postsynaptic calcium dynamics.** We modeled postsynaptic calcium concentration combining data from several experimental and theoretical sources[55,56]. Calcium ions can enter the spine via two paths, namely NMDARs and VDCCs, and quickly bind to endogenous buffers while only a small fraction remains free. Slower mechanisms, such as calcium pumps and diffusion, re-establish the intracellular calcium concentration. These dynamics for free calcium are modeled as a point current, adapting previous work due to Destexhe et al.[106], using a single ordinary differential equation (ODE) as follows

$$\frac{d}{dt}[Ca^{2+}]_i = (\tilde{I}_{NMDAR} + I_{VDCC}) \frac{\eta}{2F \cdot X} - \frac{\left([Ca^{2+}]_i - [Ca^{2+}]_i^{(0)}\right)}{\tau_{Ca}}, \tag{33}$$

where $\tilde{I}_{NMDAR}$ is the calcium component of the NMDAR-mediated current; $I_{VDCC}$ is the VDCC-mediated current; $\eta = 0.04$ is the fraction of free (non buffered) calcium as determined in Sabatini et al.[56]; $X$ is the spine volume; $F$ is the Faraday constant; $[Ca^{2+}]_i^{(0)} = 70 \times 10^{-6}\ mM$ is the intracellular calcium concentration at rest[56]; $\tau_{Ca} = 12\ ms$ is the time constant of calcium transients as determined in Sabatini et al.[56].

We validated the calcium transients generated by the model by comparing to those reported by Sabatini et al.[56] under two experimental conditions: synaptic stimulation and back-propagating AP (Supplementary Fig. A.1). We sampled a pool of synapses from L5-TTPC to L5-TTPC connections in our circuit model[5], following the acceptance criteria in Sabatini et al.[56]. In brief, we only considered synapses on basal dendrites with diameter < 2 μm, 2 <= branch order <= 4, and path length < 150 μm. Mean spine calcium transients due to repeated presynaptic activation at 0.2 Hz were found to quantitatively match the in vitro experiments (Supplementary Fig. A.1a, b). In contrast to synaptic calcium transients, we observed that AP evoked calcium transients are strongly path length dependent (Supplementary Fig. A.1c, f). As no sampling distributions are provided in Sabatini et al.[56], we assumed that our sampling of path length was more dense for proximal synapses, as synaptic voltage transients at lengths beyond 100 μm are insufficient to evoke VDCC calcium responses due to attenuation (Supplementary Fig. A.1f–h). This was implemented by restricting the pool of synapses analysed to only consider those with path length < 60 μm (Supplementary Fig. A.1f–i). Under these conditions, we found a good quantitative agreement for AP-evoked synaptic calcium transients between our model and the in vitro experiments (Supplementary Fig. A.1d, e).

The free calcium concentration described in equation (33) is filtered by a leaky calcium integrator, $c^\star$, to produce a driving signal for synaptic plasticity as follows

$$\frac{d}{dt}c^\star = -\frac{c^\star}{\tau_\star} + \left([Ca^{2+}]_i - [Ca^{2+}]_i^{(0)}\right), \tag{34}$$

where $\tau_\star$ is an appropriate time constant for calcium integration, fitted to match experimental data on LTP/LTD (see section Model fitting).

**Long-term plasticity.** Synaptic plasticity was modeled following the calcium-based formalism of Graupner and Brunel[19], which we integrated with the post-synaptic calcium transients due to NMDAR and VDCC provided by the circuit model[5]. As in Graupner and Brunel[19], the state variable $\rho$ describes the dynamics of synaptic efficacy as follows

$$\frac{d}{dt}\rho = \left(-\rho(1-\rho)(0.5-\rho) + \gamma_p(1-\rho)\Theta[c^\star - \theta_p] - \gamma_d\rho\Theta[c^\star - \theta_d]\right)/\tau, \tag{35}$$

where $\rho = 0.5$ delimits the basin of attraction of the two stable states; $\Theta$ is the Heaviside function; $\theta_d$ and $\theta_p$ are the depression and potentiation thresholds, respectively; $\gamma_d$ and $\gamma_p$ are the depression and potentiation rates, respectively; $c^\star$ is the calcium integrator described in section Postsynaptic calcium; $\tau = 70\ s$ is an appropriate time constant. The synaptic noise term, present in the original Graupner and Brunel[19] model, was removed here as we already account for the stochastic aspect of synaptic transmission and membrane potential fluctuations in the synapse and neuron models.

In our tissue model every synapse has a unique dendritic location and physiology, as its parameters are randomly drawn from appropriate distributions[5]. As a consequence, each synapse shows different calcium dynamics and no fixed thresholds could apply to all of them, even within the same connection type. For this reason, we assumed that plasticity thresholds $\theta_d$ and $\theta_p$ are expressed as linear combinations of the calcium integrator peaks during isolated presynaptic activation, $C_{pre}$, and isolated postsynaptic activation, $C_{post}$, as follows

$$\theta_d = a_{0,0} \cdot C_{pre} + a_{0,1} \cdot C_{post} \tag{36}$$

$$\theta_p = a_{1,0} \cdot C_{pre} + a_{1,1} \cdot C_{post}, \tag{37}$$

where $a_{i,j}$ are appropriate constants to be determined by the model fitting

procedure (see section Model fitting). $C_{pre}$ and $C_{post}$ are determined for each synapse numerically by running two short simulations for each connection: $C_{pre}$ is measured by a simulation where all synapses are fully activated at the same time, while $C_{post}$ by a simulation of induced post-synaptic spiking. We fit a separate set of scaling parameters for apical ($a_{i,j}$) and basal ($b_{i,j}$) synapses, as calcium dynamics and variability differed substantially between these two synapse populations. All other parameters were identical for apical and basal synapses.

**Coupling long-term plasticity and synaptic transmission models.** The peak AMPAR conductance $\hat{G}_{AMPAR}$ and the stable fixed point of the release probability $U_{SE}$ are dynamically linked to the synaptic efficacy $\rho$ of the long-term plasticity model by low pass filter dynamics:

$$\frac{d}{dt}\hat{G}_{AMPAR} = \frac{\bar{G}_{AMPAR} - \hat{G}_{AMPAR}}{\tau_{change}} \tag{38}$$

$$\bar{G}_{AMPAR} = \hat{G}_{AMPAR}^{(d)} + \rho(t) \cdot \left( \hat{G}_{AMPAR}^{(p)} - \hat{G}_{AMPAR}^{(d)} \right) \tag{39}$$

$$\frac{d}{dt}U_{SE} = \frac{\bar{U}_{SE} - U_{SE}}{\tau_{change}} \tag{40}$$

$$\bar{U}_{SE} = U_{SE}^{(d)} + \rho(t) \cdot \left( U_{SE}^{(p)} - U_{SE}^{(d)} \right), \tag{41}$$

where $\bar{G}_{AMPAR}$ is the steady-state target value of $\hat{G}_{AMPAR}$, and $\hat{G}_{AMPAR}^{(d)}$ and $\hat{G}_{AMPAR}^{(p)}$ are the conductance values of the depressed and potentiated states, respectively; $\bar{U}_{SE}$ is the steady-state target value of $U_{SE}$, and $U_{SE}^{(d)}$ and $U_{SE}^{(p)}$ are the release probabilities of the depressed and potentiated states, respectively. As the synaptic efficacy, $\rho$, is bistable[19], so too are $\hat{G}_{AMPAR}$ and $\bar{U}_{SE}$ to which $\hat{G}_{AMPAR}$ and $U_{SE}$ relax with a time constant $\tau_{change} = 100$ s to approximate the typical expression time course observed in vitro[12,14]. The decision to make $\hat{G}_{AMPAR}$ and $U_{SE}$ dynamic variables, as opposed to an instantaneous mapping as in Graupner and Brunel[19], is motivated by the observation that the expression of synaptic plasticity is known to be slow compared to its induction[12,14].

**Long-term plasticity parameters initialization.** We assigned the initial state of the synapse $\rho_0 \equiv \rho(t=0)$ to the potentiated ($\rho_0 = 1$) or depressed ($\rho_0 = 0$) state at random, using the individual synapse release probability, $U_{SE}$, prescribed in the circuit model[5] as the probability of being in the potentiated state. This initialization strategy reconciled the synaptic transmission and plasticity parameters, preventing unreasonable configurations, such as fully potentiated connections with very small release probabilities. As synaptic parameters are strongly correlated (see section Correlation of synaptic parameters), this choice favours strong synapses to be initialized in the potentiated state, and weaker synapses to be assigned to the depressed state.

We initialized the conductance values of the depressed and potentiated states as follows

$$\hat{G}_{AMPAR}^{(d)} = \begin{cases} \hat{G}_{AMPAR}^{(0)} & \text{if } \rho_0 = 0 \\ \frac{1}{2}\hat{G}_{AMPAR}^{(0)} & \text{if } \rho_0 = 1 \end{cases} \tag{42}$$

$$\hat{G}_{AMPAR}^{(p)} = \begin{cases} 2\hat{G}_{AMPAR}^{(0)} & \text{if } \rho_0 = 0 \\ \hat{G}_{AMPAR}^{(0)} & \text{if } \rho_0 = 1, \end{cases} \tag{43}$$

where $\hat{G}_{AMPAR}^{(0)}$ is the initial conductance value prescribed in the circuit model[5]. The potentiation/depression ratio of the conductance is based on the observation in the hippocampus that $\hat{G}_{AMPAR}^{(p)} \approx 2\hat{G}_{AMPAR}^{(d)}$[89].

We assumed an exponential relationship between the potentiated and depressed state to account for release probability saturation:

$$U_{SE}^{(p)} = \left( U_{SE}^{(d)} \right)^{\nu}, \tag{44}$$

where $\nu$ is an appropriate constant. From the work of Enoki et al.[90] we estimated the value of $\nu$ to match the set of experiments where strong and repeated LTP/LTD was induced. We found that $\nu \in (0.1, 0.25)$ would provide a satisfactory approximation and chose $\nu = 0.2$ for simplicity. We then initialized $U_{SE}^{(d)}$ and $U_{SE}^{(p)}$ at every synapse as follows

$$U_{SE}^{(d)} = \begin{cases} U_{SE}^{(0)} & \text{if } \rho_0 = 0 \\ \sqrt[\nu]{U_{SE}^{(0)}} & \text{if } \rho_0 = 1 \end{cases} \tag{45}$$

$$U_{SE}^{(p)} = \begin{cases} \left( U_{SE}^{(0)} \right)^{\nu} & \text{if } \rho_0 = 0 \\ U_{SE}^{(0)} & \text{if } \rho_0 = 1, \end{cases} \tag{46}$$

where $U_{SE}^{(0)}$ is the initial release probability prescribed in the circuit model[5].

**Correlation of synaptic parameters.** It is well established that several morphological and physiological variables of excitatory synapses are correlated. For example, several studies have shown that postsynaptic density (PSD) area is strongly correlated with pre- and post-synaptic variables, such as spine head volume[57–59], bouton volume and number of vesicles[57]. Such findings suggest that a certain degree of parameter correlation is required to accurately model variability in synaptic transmission and post-synaptic calcium dynamics.

Based on the available experimental evidence, we imposed the following correlations to synaptic model variables. The relationship between the total number of vesicles and PSD area or spine volume, $X$, was estimated in Harris and Stevens[57]. Assuming the total number of vesicles is proportional to the size of the RRP, $N_{sites}$, as is the PSD area to conductance, we could impose the experimentally observed correlation coefficients as follows

$$\rho(\hat{G}_{AMPAR}, N_{sites}) = 0.9 \tag{47}$$

$$\rho(X, N_{sites}) = 0.92. \tag{48}$$

We set the correlation between spine volume and synaptic conductance based on equivalent measurements from Harris and Stevens[57] (hippocampus) and Arellano et al.[59] (neocortex) as follows

$$\rho(X, \hat{G}_{AMPAR}) = 0.88. \tag{49}$$

Accounting for this correlation is important because it allows us to predict spine volumes, an unknown variable, from conductance, a parameter prescribed by the tissue model[5] (see section Dendritic spines). Unfortunately, we are not aware of any report explicitly quantifying the correlation between release probability and synaptic conductance. As these two variables will evolve to become correlated due to the synaptic plasticity model, we assumed also a high initial correlation as follows

$$\rho(U_{SE}, \hat{G}_{AMPAR}) = 0.9. \tag{50}$$

Based on the above prescribed correlation coefficients, we populated a correlation matrix, and filled the missing entries using the simple algorithm proposed by Kahl and Günther[107], obtaining the final correlation matrix $M$ used in this work for the vector of parameters $P$:

$$P = \begin{bmatrix} U_{SE} & N_{sites} & \hat{G}_{AMPAR} & X \end{bmatrix} \tag{51}$$

$$M = \begin{bmatrix} 1 & 0.81 & 0.9 & 0.79 \\ 0.81 & 1 & 0.9 & 0.92 \\ 0.9 & 0.9 & 1 & 0.88 \\ 0.79 & 0.92 & 0.88 & 1 \end{bmatrix} \tag{52}$$

The synaptic parameters in $P$ in the circuit model were then re-sampled from a multi-variate normal distribution with covariance matrix $M$ and remapped to the corresponding marginal distributions prescribed in the circuit model. Note, after determining $X$ according to this method, $\hat{G}_{VDCC}$ is calculated from $X$ using Equation (28). Parameters of individual synapses belonging to the same connection are sampled independently of each other.

**Simulations and data analysis.** All in silico experiments were performed using NEURON[108] and the Blue Brain Project (BBP) tissue model[5]. Analysis routines were written in Python, and make use of standard scientific packages: Matplotlib[109], SciPy[110], NumPy[111], Pandas[112], Seaborn[113], Jupyter[114]. Neuron models from the tissue model were modified to remove a calcium-activated potassium conductance from the cell body and axon initial segment. This change was required to eliminate extreme hyper-polarization during high frequency stimulation which prevented plasticity model fitting, and is an artefact expected to be corrected in future releases of the tissue model[5]. Since in NEURON synapses are considered postsynaptic processes activated by a presynaptic trigger, we did not need to simulate the presynaptic cell during our experiments and we would not obtain any benefit in doing so. Rather we computed the desired spike timing and fed it to the synaptic processes, hosted on the postsynaptic cell. To further reduce the computational cost of each simulation, we fast-forwarded the convergence of the LTP/LTD model variables after the induction. That is, after establishing whether stimulation was sufficient to cause a state change, we moved the interested synaptic variables to their new fixed points rather than simulating their (slow) convergence. This trick was used during model fitting to substantially reduce the computational cost, but was disabled for experiments showing the full convergence dynamics. Fast-forwarding the convergence dynamics does not affect plasticity outcomes in any way, as long-term plasticity outcomes are defined here from steady state values. Calcium thresholds cannot be crossed during the fast-forward process by definition, as they are always higher than the transients generated by sparse pre- and post-synaptic stimulation (see section Model fitting).

**Statistics.** Statistical analysis was performed using Python/SciPy (Welch's unequal variances t-test, two-sided) and R/ks[115] (kernel density estimate (KDE) test, see also R/fasano.franceschini.test for an alternative implementation[116]).

**Reproducing in vitro experiments in silico**. To ensure comparability and correspondence between experimental data and in silico simulations, we reproduced also the typical biases of multi patch-clamp in vitro experiments. In particular, neurons were randomly selected mimicking the tendency of experimenters to patch nearby cells on the same focal plane (e.g. a $50 \times 50 \times 10$ μm volume for layer 5 thick-tufted pyramidal cell (TTPC) connections). This typical bias is motivated by the desire to maximize connection probability in lab experiments and thereby slice yield. The specifics of individual experiments for each connection type considered in this work are described below.

L5-TTPC to L5-TTPC: Connections were selected from random volumes of $50 \times 50 \times 10$ μm following the methods in Markram et al.[12], Sjöström et al.[14], Sjöström and Häusser[36]. Stimulation protocols and data analysis could be reproduced as described.

L2/3-PC to L5-TTPC: Connections were selected from random volumes of $50 \times 700 \times 10$ μm, following the methods in Sjöström and Häusser[36]. Simulation protocols and data analysis could be reproduced as described.

L4-PC to L2/3-PC: Connections were selected from random volumes of $50 \times$ max $\times 10$ μm, following the methods in Rodríguez-Moreno and Paulsen[42]. Stimulation protocols and data analysis could be reproduced as described. Note, we computed excitatory post-synaptic potential (EPSP) slopes from the average traces before and after, which may differ from the purported approach taken in Rodríguez-Moreno and Paulsen[42] of averaging the slope of individual sweeps. We emulated the effects of MK801 by setting the LTD rate $\gamma_d = 0$.

L2/3-PC to L2/3-PC: Connections were selected from random volumes of $50 \times 50 \times 10$ μm, following the methods in Egger et al.[41], Banerjee et al.[43], Zilberter et al.[44]. In the Egger et al.[41] experiments, the specific somatosensory cortex region of the in vitro experiments was different from the one of our tissue mode (barrel cortex in Egger et al.[41], non-barrel cortex in Markram et al.[5]). In the Banerjee et al.[43] experiments, the EPSP slope was used to compute EPSP ratios. Data analysis was performed as described in Egger et al.[41], Banerjee et al.[43], Zilberter et al.[44], with the exception that the Gaussian weighting method in Egger et al.[41], which could not be applied to our data. This technique, used to reduce the error on the mean EPSP ratio estimate, requires data to be normally distributed. This condition is not met for every protocol in our data after plasticity induction.

L4-SSC to L4-SSC: Connections were selected from random volumes of $50 \times 50 \times 10$ μm, following the methods in Egger et al.[41]. Data analysis was performed with the same considerations as for L2/3-PC to L2/3-PC.

L5-TTPC to L5-TTPC in low calcium: Connections were selected from random volumes of $50 \times 50 \times 10$ μm, following the methods in Markram et al.[12], Sjöström and Häusser[36]. We model the low calcium conditions in vivo by (a) reducing the synaptic release probability to 15% of its in vitro value reflecting $[Ca^{2+}]_o = 1.2$ mM as described in Markram et al.[5]; (b) adapting the calcium reversal potential based on the Nernst equation; (c) recomputing the fractional component of calcium current through the NMDARs using equation (23), and (d) recomputing the calcium threshold scaling factors under the new conditions, i.e. $C_{pre}$ and $C_{post}$, which are determined by isolated pre- and post-synaptic activity, respectively.

**Model fitting**. To fit the 11 free parameters (8 threshold parameters, $\gamma_p$, $\gamma_d$, and $\tau_\star$) of the plasticity model, we reproduced in vitro experiments from Markram et al.[12], Sjöström and Häusser[36] using our in silico model of cortical tissue, as described in section Reproducing in vitro experiments in silico. We then used a multi-objective genetic algorithm (GA)[65,66] to find model parameters that provide the best match between in silico and in vitro mean EPSP ratios, respectively. The GA was run for a total of 103 generations. After 25 generations of the GA, a chimera solution, obtained by cloning and continuing the optimization for 76 generations using a surrogate evaluation function, was injected to the GA population in an attempt to speed up convergence. The surrogate evaluation function was obtained by fitting a boosted tree regression model[117] which maps the model parameters to the expected fitness. To minimize the computational cost, we considered only 5 stimulation protocols as targets for the optimization, as summarized in Table 1. The best solution was taken as the individual over all generations in the optimization history minimizing the aggregated error, defined as the maximum of its errors across the protocols in the training set, and is reported in Table 2. The error on each protocol is defined as follows

$$\text{error} = \frac{|\bar{R}_{\text{insilico}} - \bar{R}_{\text{invitro}}|}{\text{Standard Error}\left(\bar{R}_{\text{invitro}}\right)}, \quad (53)$$

where $\bar{R}_{\text{insilico}}$ and $\bar{R}_{\text{invitro}}$ are the mean EPSP ratios of the in silico experiments and target in vitro experiments, respectively. The plasticity model was optimized using the Blue Brain 5 supercomputer https://www.cscs.ch/computers/blue-brain-5, hosted at the Centro Svizzero di Calcolo Scientifico (CSCS) in Lugano, Switzerland. The optimization procedure employed checkpointing to provide fault tolerance and facilitate the scheduling of work on the shared compute resource. For reference, a single generation of the GA required approximately 20480 core-hours on Blue Brain 5.

We assessed model generalization on a held-out out set of in vitro experiments from Egger et al.[41], Rodríguez-Moreno and Paulsen[42], Banerjee et al.[43], Zilberter

**Table 2 Optimized parameters with respective boundaries and best solution for the long term plasticity model.**

| Parameter | Bounds | Best | Description |
|---|---|---|---|
| $\tau_\star$ | (150, 350) s | 278.318 | Time constant of calcium integrator |
| $a_{0,0}$ | $(1, 10)^\dagger$ | 1.127 | $C_{pre}$ factor for $\theta_d$ apical |
| $a_{0,1}$ | (1, 5) | 2.456 | $C_{post}$ factor for $\theta_d$ apical |
| $a_{1,0}$ | $(1, 10)^\dagger$ | 5.236 | $C_{pre}$ factor for $\theta_p$ apical |
| $a_{1,1}$ | (1, 5) | 1.782 | $C_{post}$ factor for $\theta_p$ apical |
| $b_{0,0}$ | (1, 5) | 1.002 | $C_{pre}$ factor for $\theta_d$ basal |
| $b_{0,1}$ | (1, 5) | 1.954 | $C_{post}$ factor for $\theta_d$ basal |
| $b_{1,0}$ | (1, 5) | 1.159 | $C_{pre}$ factor for $\theta_p$ basal |
| $b_{1,1}$ | (1, 5) | 2.483 | $C_{post}$ factor for $\theta_p$ basal |
| $\gamma_d$ | (1, 300) | 101.5 | Depression rate |
| $\gamma_p$ | (1, 300) | 216.2 | Potentiation rate |

$^\dagger$ Extended to (1, 15) after generation 25 of the optimization.

et al.[44], Sjöström et al.[14], i.e. data that were not previously used during the model fitting process.

**Visualization**. The cortical column visualization in Fig. 1 was created with Brayns[118]. The neuronal renderings in Figs. 3, 4, 5 and 7 were generated with NeuroMorphoVis[119].

**Reporting summary**. Further information on research design is available in the Nature Research Reporting Summary linked to this article.

## Data availability
The in silico EPSP measurements, plasticity ratios and optimization results are publicly available on Zenodo at https://doi.org/10.5281/zenodo.5654788[49].

## Code availability
We made the synapse model, simulations and analysis code used in this work publicly available on Zenodo at https://doi.org/10.5281/zenodo.5654788[49]. Simulations can be reproduced using EModelRunner https://github.com/BlueBrain/EModelRunner, a open source Python package developed to run cell models provided by the Blue Brain Project portals.

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

## Acknowledgements

We thank Michael Hines for helping with synapse model implementation in NEURON; Mariana Vargas-Caballero for sharing NMDAR data; Veronica Egger for sharing in vitro data and for clarifications on the analysis methods; Jesper Sjöström for sharing in vitro data, helpful discussions, and feedback on the manuscript; Ralf Schneggenburger for helpful discussions and clarifications on the NMDAR calcium current model; Fabien Delalondre for helpful discussions; Francesco Casalegno and Taylor Newton for helpful discussion on model fitting; Daniel Keller for helpful discussions on the biophysics of synaptic plasticity; Natali Barros-Zulaica for helpful discussions on MVR modeling and generalization; Srikanth Ramaswamy, Michael Reimann and Max Nolte for feedback on the manuscript; Wulfram Gerstner and Guillaume Bellec for helpful discussions on synaptic plasticity modeling. This study was supported by funding to the Blue Brain Project, a research center of the École polytechnique fédérale de Lausanne, from the Swiss government's ETH Board of the Swiss Federal Institutes of Technology. E.B.M. received additional support from the CHU Sainte-Justine Research Center (CHUSJRC), the Institute for Data Valorization (IVADO), Fonds de Recherche du Québec–Santé (FRQS), the Canada CIFAR AI Chairs Program, the Quebec Institute for Artificial Intelligence (Mila), and Google. R.B.P. and J.DF. received support from the Spanish "Ministerio de Ciencia e Innovación" (grant PGC2018-094307-B-I00). M.D. and I.S. were supported by a grant from the ETH domain for the Blue Brain Project, the Gatsby Charitable Foundation, and the Drahi Family Foundation.

## Author contributions

G.C., H.M., and E.B.M. designed the study. G.C., M.G., and E.B.M. designed the plasticity model. V.D. contributed to designing initial versions of the plasticity model. C.R. designed and parameterized the calcium and VDCC models. C.R. and R.B.P. acquired, analyzed, and curated the spine volume dataset. G.C. and C.R. designed the spine volume prediction algorithm. G.C., M.D., and O.A. extended and parameterized the NMDAR model. G.C., J.K., and P.K. designed and curated the NEURON implementation of the synapse model. G.C. ran the simulations, and the model parameters optimization, and performed the data analysis. A.E. contributed to the NEURON implementation of the synapse model, to the running of simulations, and to data analysis. E.B.M. contributed to the validation of the calcium and VDCC models, and to data analysis. G.C. and R.P. adapted the in silico protocols and methods to reproduce in vitro experiments. G.C. and W.V.G. designed the model parameters optimization method. A.T.J. and A.M.T. developed the BBP pairs exporter and the software needed to run pair simulations. M.A. and C.M. produced the neuron renderings from the BBP circuit model and the illustrations for the paper. M.G., H.M., and E.B.M. supervised the research. J.D.F. and I.S. contributed to supervising the research. G.C. and E.M. wrote the paper. H.M., M.G., and I.S. contributed to writing the paper.

## Competing interests

The authors declare no competing interests.
