## [Peer review file · Nature Communications]

REVIEWER COMMENTS

Reviewer #1 (Remarks to the Author):

This study of Chindemi et al. implements a modified version of a previously developed plasticity model (Graupner and Brunel, 2012) to investigate the hypothesis that PC type variability (seen as variability in calcium dynamics) and a single plasticity rule (with fixed parameters), can be generalized to explain multiple plasticity phenomena. Towards this goal, after optimizing the parameters to specific plasticity-inducing protocols for the L5-L5 and L2/3-L5 connections, they test the generalization to other protocols/types of connections, using the model of the neocortical microcircuit. The plasticity model is also used to predict responses under in vivo like conditions (physiological extracellular calcium concentrations). This work is well written, methodological robust and builds upon previous models published by the authors. Yet, the novelty of the work is confined by previously published models that have shown similar phenomena (e.g. Graupner and Brunel, 2012, Fig. 2) and the lack of statistical analysis. Following are the comments, not in order of appearance in the text:

Major Comments:

1. There is no statistical test to evaluate the differences (or not) between the model and the in vitro experiments, whenever experimental data are available (apart from the training set, Fig. 2c). As the main claim of the authors is that this rule can generalize to multiple protocols/types of connections, the lack of statistical comparisons between the simulated and in vitro data makes this claim unsubstantiated.
2. Graupner et al., systematically explored the dependence of plasticity as a function of the calcium amplitude. Although I understand that the authors are focusing on the biologically relevant implementation of this rule (that is, the calcium dynamics observed in the biologically realistic microcircuit model), which is/are the main parameter(s) that determine the different responses among the different types of connections? Is it, for example, the spine volume distribution?
3. As the parameters used for the apical and basal trees are different, how does the implemented plasticity rule, with the optimized parameters, shape the respective EPSPs in the different simulated connections/protocols? This would produce interesting predictions regarding the apical versus basal tree plasticity mechanisms (as the authors also point out in the discussion, lines 392-397).
4. The authors do not model the pre-synaptic/postsynaptic changes over time (line 362) which means their model ignores the constraints imposed by other phenomena, such as the synaptic competition (<https://www.pnas.org/content/111/33/12217>) during LTP consolidation. These phenomena may lead to very different LTP outcomes from the ones that the model predicts. The authors could expand on the consequences of LTP consolidation phase on the predictions of their model.

Minor Comments:

There are some inconsistencies in the parameter names between results and materials and methods. E.g. equation 4,5 and 38-41 are the same but with different parameter names, and τ^* in page 7 and τ_{effcai} in Table 3.

Table 3: Time constant of calcium integrator, please delete the 'of' repetition.

Reviewer #2 (Remarks to the Author):

The authors use an extension of their previously published synaptic plasticity learning rule to explain experimentally published synaptic plasticity for four types of pyramidal cell connections in the neocortex. Integrating pre-synaptic factors, such as short term plasticity of vesicle release, with post-synaptic factors and a phenomenological model of calcium dynamics is quite novel. The authors use a multi-objective evolutionary strategy to find 11 parameters that will fit the synaptic plasticity results of a set of experiments, and then show that the model also does a fair job of matching synaptic plasticity results in additional sets of experiments. The most exciting part of the research is the ability to fit diverse sets of synaptic plasticity experiments, spanning several neuron types, with a single set of parameters.

I have several concerns about the manuscript as submitted.

1. The predictions and fit to data in Figure 3 are quite nice. Interestingly, Fig 3f show LTD using 50 ms ISI. Given that this is a known issue of plasticity rules (erroneously predicting LTD for long positive ISI), it is important to show experimental support of LTD using this protocol.

2. The authors should report on how many parameter sets fit the test set and how many parameter sets did not fit. The prediction of plasticity outcomes is the most exciting part of the research; however I was disappointed to see that none of the predictions were experimentally tested. All of the novel comparisons, even beyond the generalizations, were to data that had been previously published. Thus, it is unclear whether these are true predictions, since multi-objective optimization provides multiple parameter sets. Were there other parameter sets that didn't correctly generalize? If there were multiple parameter sets that would correctly generalize, it would strengthen the manuscript to show that the generalization was robust to those parameters. On the other hand, the authors discuss some data sets that were excluded because the experiments were from the wrong species, brain region or age, e.g. "we had experimental data for only one connection type and protocol featuring mostly apical synapses, and it was not from somatosensory cortex but rather the visual cortex." This seems to contradict an earlier part of the paragraph which states "we included data from rat visual cortex and from mouse somatosensory cortex" - clearly from two brain regions and two species. A more convincing argument needs to be made that the data sets were not selected based on the ability of the model to fit the data.

3. The code is not provided. Neither the optimization code, nor the post-synaptic neuron models and synapse locations used for the simulations. The authors provide a URL for parameters, but this is not sufficient for reproducibility and it is unclear whether the rest of the code is part of the internal release and how to run the simulations. It is unclear until the very end of the manuscript if the entire column was simulated, or just the single post-synaptic neurons. If the latter, it seems that the purpose of the entire column was to determine connectivity. Line 678 indeed explains that the entire column was not simulated, but this needs to be stated much earlier in the manuscript. All the code used to run parameter optimizations and simulations of synaptic plasticity must be made available. Also, a table of parameters for equations provided in the text should be provided. The reader should not have to wade through the entire blue brain project trying to find model parameters.

4. Calcium channel conductance and volume used in Equation 1 and described on lines 118-119 are unclear. The additional explanation in methods lines 572-573 does not help. The authors should state explicitly that there are no spines in this model. In addition, the explanation of spine volume seems circular since you determined spine volume from conductance, and then 2 sentences down in the methods stated that synaptic VDCC conductance used spine head dimensions. You mention a prescribed conductance distribution, which suggests it has been specified, but I don't see where the conductance distribution is specified. Equally important, what are the consequences for the plasticity rule of not making spine head volume proportional to calcium conductance? I see that correlation between spine head volume and AMPA conductance is finally explained in equation 51&52, but this does not include VDCC conductance.

5. The authors claim their calcium model is constrained by experimental data (lines 140-141), but do not reference calcium imaging publications that measure spine calcium during STDP or high frequency stimulation. Nor do they compare their calcium dynamics with other experimental measurements of calcium dynamics. The authors should provide direct experimental comparison in Figure 1 together with references used to constrain the calcium equation parameters.

Minor comments.

In several places, including the discussion, the authors refer to phenomenological models as being insufficient for their purposes. But in fact this is a phenomenological model. The authors do not model diffusion, explicit buffers and pumps. Thus, it is a bit of a stretch to state (in the discussion) that "the model takes advantage of the full calcium time course", especially since the model requires an additional ad hoc calcium integrator equation. Similarly, the bistable synaptic plasticity equation is phenomenological.

line 229-230: It is unclear how being able to fit the synaptic plasticity data "supports our assumptions on the initial ratio of potentiated over depressed synapses, and the conductance and release probabilities of the depressed and potentiated states."

line 261: It is unclear how this is valid. Clearly, setting the ltd induction rate to 0 will block LTD, but it is unclear how this is comparable to blocking presynaptic NMDAR. Modifying the pre-synaptic equations would be more appropriate.

line 318: The authors actually present data on plasticity that depends on pre-syn NMDA receptors, which the authors model cannot implement exactly. This data shows that the plasticity mechanisms are not shared and contradicts the authors claims.

Line 481 and Eqn 15-16: Since this is describing the synaptic response to vesicle release, and you have distinct equations for the pre-synaptic side, why is N_{sites} part of this equation?

lines 499-516 are very confusing. 1st, why provide and explain an equation you then state you are not using. 2nd, at what values is it undefined? Doesn't that depend on Ca , M , pm and pca , which I couldn't find in the manuscript? Why not use the GHK flux equation which does not normalize to all other currents, and which is well defined (using L'Hospital's Rule).

This statement is unclear: "We assigned the initial state of the synapse to the potentiated ($pho0 = 1$) or depressed ($pho0 = 0$) state at random, using the individual synapse release probability, U_{SE} ". Please provide additional information.

Eqn 36,37 are not complete. The term C_{pre} is not defined. It seems this is left over from the previously published model. Given that their are explicitly equations for vesicle release (lacking in the previously published model), this term should not be needed.

Eqn 38-42 are incomplete. g_{max} depends on g_d and g_p , but g_d and g_p depend on initial $g_{max,0}$, which is not defined in this section.

Eqn 45-46 have a term x in them which is not defined, possibly that should be γ .

line 673: I don't know what hot-fixed means.

Figures A.2-A.11 have no y axis labels and none of figure captions explain the graphs' meaning.

Reviewer #3 (Remarks to the Author):

The manuscript presents a model of long-term synaptic plasticity based on the dynamics of postsynaptic calcium. The novelty of this approach is that a single calcium-based mechanism is sufficient to reproduce experimental data from a number of synapses between different cortical cell types, with observed differences arising from other properties of the postsynaptic cells such as overall morphology and synapse location. The model combines detailed biophysical modelling of voltages and calcium currents with phenomenological elements to account for the integration of calcium signals on relatively long timescales and a bistable synaptic efficacy model introduced in Graupner and Brunel (2012) to allow for plasticity. The claims are novel in that a single model that can predict both depression and potentiation between detailed models of different cortical cell types, and potentially at different (in vivo) calcium concentrations, did not previously exist and the paper is likely to be of interest to the community. The stated intent of the paper is to provide a null model for cortical excitatory plasticity against which other models and data can be judged and I believe that this is achieved; the results are likely to be a useful comparison to for future plasticity models with additional details or in different brain regions.

The models and methods of analysis are well described and build on a number of techniques described in previous papers from the Blue Brain Project. The implementation of the plasticity model is likely to be reproducible for single neurons by independent researchers, but the level of computational power required to reproduce the full set of results in the paper would be prohibitive for many.

I have a couple of more detailed comments:

Line 173 (etc): The correlation of (plasticity in) synaptic conductance and release probability is a strong assumption and is not entirely supported by the cited experimental literature, which only explicitly finds this relationship in hippocampal synapses. It is not unreasonable to make this assumption, but it does restrict the range of synaptic dynamics that can emerge, particularly given the importance of stochasticity in synaptic transmission and ongoing plasticity.

Lines 370, 537 (etc): The fact that the number of vesicle release sites is fixed in the model is another feature that will influence the statistics of potentiated and depressed synapses. The authors do discuss the reasons for focussing on AMPAR-mediated plasticity as the determinant of EPSP amplitude in the Discussion, but it might also be interesting to comment on the implications of relaxing this restriction.

Eqs 1, 13, 44: The symbol gamma is used to represent three different constants, it might be clearer to use different symbols.

Line 126: Typo 'vesicles'.

Point-by-point response to the reviewers' comments

Reviewer #1 (Remarks to the Author)

This study of Chindemi et al. implements a modified version of a previously developed plasticity model (Graupner and Brunel, 2012) to investigate the hypothesis that PC type variability (seen as variability in calcium dynamics) and a single plasticity rule (with fixed parameters), can be generalized to explain multiple plasticity phenomena. Towards this goal, after optimizing the parameters to specific plasticity-inducing protocols for the L5-L5 and L2/3-L5 connections, they test the generalization to other protocols/types of connections, using the model of the neocortical microcircuit. The plasticity model is also used to predict responses under in vivo like conditions (physiological extracellular calcium concentrations). This work is well written, methodological robust and builds upon previous models published by the authors. Yet, the novelty of the work is confined by previously published models that have shown similar phenomena (e.g. Graupner and Brunel, 2012, Fig. 2) and the lack of statistical analysis. Following are the comments, not in order of appearance in the text: Major Comments:

1. There is no statistical test to evaluate the differences (or not) between the model and the in vitro experiments, whenever experimental data are available (apart from the training set, Fig. 2c). As the main claim of the authors is that this rule can generalize to multiple protocols/types of connections, the lack of statistical comparisons between the simulated and in vitro data makes this claim unsubstantiated.

We thank the reviewer for raising this important issue. We revised the manuscript to include statistical testing wherever possible. We used the Welch's unequal variances t-test to compare the predictions of the model against the in vitro experimental results. We could not find any significant difference for all tested conditions except for a few cases: the L23-PC to L23-PC experiment in Figure 4c; L4-SSC to L4-SSC experiments in Supplementary Figure A.7; some protocols in the visual cortex for L5-TTPC to L5-TTPC and L2/3-PC to L2/3 PC in Supplementary Figure A.4 and A.6. These exceptions are now discussed in detail in the main text. Briefly, (1) for the L23PC to L23PC protocol in Figure 4c our model correctly predicts LTP, but of a stronger magnitude than was observed; (2) for the L4-SSC to L4-SSC experiments in Supplementary Figure A.7, the mismatch confirms the specificity of our model because plasticity of spiny stellate cells is reported to be mediated by different mechanisms than the pyramidal NMDAR-mediated plasticity considered here (see main text; Egger et al. 1999); and (3) some degree of region-specific differences between a model based on somatosensory cortex and experimental data from the visual cortex are to be expected. We added the results of the corresponding t-tests (p-value) to the caption of each figure where appropriate. Furthermore, we now use the same statistical test to compare model predictions in high and low calcium conditions in Figure 7. We updated the analysis in Figure 2c to also rely on this test, which is more appropriate than the one-sample t-test previously used). This last change did not alter the previously reported results (i.e. comparisons remained non significant).

2. Graupner et al., systematically explored the dependence of plasticity as a function of the calcium amplitude. Although I understand that the authors are focusing on the biologically relevant implementation of this rule (that is, the calcium dynamics observed in the biologically realistic

microcircuit model), which is/are the main parameter(s) that determine the different responses among the different types of connections? Is it, for example, the spine volume distribution?

Thanks to the reviewer for this insightful comment. To investigate the role of synaptic parameters in connection-type specific STDP, we performed additional analysis of our simulations and models. In brief, we found that synaptic conductance and apical ratio exhibit a connection-type specificity that could account for the different STDP types observed in our simulations (Figure 6). The text has been revised to include a new sub-section "Connection-type specific STDP" specifically devoted to answer this question, and two additional supplementary figures (Supplementary Figure A.8 and A.9) have been added.

3. As the parameters used for the apical and basal trees are different, how does the implemented plasticity rule, with the optimized parameters, shape the respective EPSPs in the different simulated connections/protocols? This would produce interesting predictions regarding the apical versus basal tree plasticity mechanisms (as the authors also point out in the discussion, lines 392-397).

Thanks for raising this interesting question. In responding to the related previous comment of the reviewer (comment #2), we performed a broad analysis of the dependence of plasticity on synaptic parameters & innervation features, such as apical ratio. This analysis also reveals an answer to this specific question. In particular, we found that plasticity outcomes of specific connection types and protocols could be clustered according to apical ratio and NMDAR conductance. Moreover, a larger apical ratio (such as for the L2/3 PC to L5 TTPC connection type) results in a shift towards LTD and a reduction in sensitivity to spike-timing, but not frequency. We discussed these observations in the main text and updated the discussion accordingly. The analysis is shown in the two additional supplementary figures (Figure X and Y).

4. The authors do not model the pre-synaptic/postsynaptic changes over time (line 362) which means their model ignores the constraints imposed by other phenomena, such as the synaptic competition (<https://www.pnas.org/content/111/33/12217>) during LTP consolidation. These phenomena may lead to very different LTP outcomes from the ones that the model predicts. The authors could expand on the consequences of LTP consolidation phase on the predictions of their model.

While we do model the dynamics of early LTP/LTD, the reviewer raises an important point that LTP/LTD undergo changes beyond the early phase that are important for learning outcomes, and which we do not account for in this study. It would be possible, in principle, to extend the model to include consolidation dynamics following e.g. Clopath et al., 2008. However, the focus of the present study is on modeling experimental data on synaptic plasticity in a connection-type specific manner. Accounting for consolidation dynamics would face the problem that there is currently insufficient data on connection-type specificity in the neocortex, and so it is beyond the scope of the current study. We revised line 362 to clarify that we do indeed account for the dynamics of LTP/LTD expression, and revised the discussion to include a paragraph covering plasticity consolidation.

Minor Comments:

There are some inconsistencies in the parameter names between results and materials and methods. E.g. equation 4,5 and 38-41 are the same but with different parameter names, and τ^* in page 7 and $\tau_{\text{eff}}^{\text{cai}}$ in Table 3.

Table 3: Time constant of of calcium integrator, please delete the 'of' repetition.

We thank the reviewer for pointing out these mismatches. We removed all inconsistencies between the Results, Methods and Tables sections. The typo in Table 3 (Table 2 in the revised manuscript) was fixed as well.

Reviewer #2 (Remarks to the Author)

The authors use an extension of their previously published synaptic plasticity learning rule to explain experimentally published synaptic plasticity for four types of pyramidal cell connections in the neocortex. Integrating pre-synaptic factors, such as short term plasticity of vesicle release, with post-synaptic factors and a phenomenological model of calcium dynamics is quite novel. The authors use a multi-objective evolutionary strategy to find 11 parameters that will fit the synaptic plasticity results of a set of experiments, and then show that the model also does a fair job of matching synaptic plasticity results in additional sets of experiments. The most exciting part of the research is the ability to fit diverse sets of synaptic plasticity experiments, spanning several neuron types, with a single set of parameters.

I have several concerns about the manuscript as submitted.

1. The predictions and fit to data in Figure 3 are quite nice. Interestingly, Fig 3f show LTD using 50 ms ISI. Given that this is a known issue of plasticity rules (erroneously predicting LTD for long positive ISI), it is important to show experimental support of LTD using this protocol.

Thanks to the reviewer for this perceptive observation. We have revised the results section to include a paragraph discussing this finding, and its support by previous experiments in the literature. Specifically, Sjöström et al., 2001 reported a flip from LTP to LTD at $\Delta t = 25$ ms, freq. = 20 Hz (see Figure 7C), a configuration where the pre- and post-synaptic spike trains are perfectly interleaved (i.e. the ISI of the pre- and post-synaptic spike trains are $2 \times \Delta t$). This experimental finding supports the flip from LTP to LTD around $\Delta t = 50$ ms for freq. = 10 Hz predicted by our model, similarly a configuration where pre- and post-synaptic spike trains are perfectly interleaved. Given the short ISI of the spike trains in these protocols, we restrict our analysis to $\Delta t \leq \text{ISI}/2$, as larger Δt shifts the closest matching spike from pre-post to post-pre or vice versa.

2. The authors should report on how many parameter sets fit the test set and how many parameter sets did not fit. The prediction of plasticity outcomes is the most exciting part of the research; however I was disappointed to see that none of the predictions were experimentally tested. All of the novel comparisons, even beyond the generalizations, were to data that had been previously published. Thus, it is unclear whether these are true predictions, since multi-objective optimization provides multiple parameter sets. Were there other parameter sets that didn't correctly generalize? If there were multiple parameter sets that would correctly generalize, it would strengthen the manuscript to show that the generalization was robust to those parameters.

As pointed out by the reviewer, our optimization algorithm produces multiple parameter sets that, in principle, could have very different generalization properties. However, we found that the best solutions produced by our optimization algorithm were all clustered in parameter space, with a gradual degradation of performance towards the perimeter of the cluster. This indicates that the optimization has converged to a smooth local minimum. Furthermore, the overall best solution was selected using an objective metric (i.e. the solution minimizing the maximum error across the training protocols), chosen a priori and independent of the test set performance. In light of these new analyses, we don't expect that spending further computational resources to assess generalization performance on the test set for the best solutions in our pool (e.g. top 5%) will reveal significant differences, given how homogenous

this group of solutions would be in terms of parameters. We added a paragraph in the results and methods sections to describe the new analyses and the selection process for the best solution, and two novel figures to illustrate the distributions of errors and parameters around the best solution (Supplementary Figure A.2 and A.3).

The reviewer's comment implicitly raises a more fundamental discussion point, that of using the test set to make model selection or design choices during model development, which will lead to an overestimation of the final model generalization performance. This issue is an example of what is generally referred to as data leakage (or train-test contamination).

We are aware of this matter and followed best practices to avoid it. For example, in the initial stages of this work, the L2/3 to L5 connection type was the only data included in the test set. However, we were unable to get early versions of the model to generalize to this connection type under the hypothesis of a single parameter set for both apical and basal synapses, and based on this we made a fundamental revision of the model to include separate parameters for basal and apical synapses. As this decision was made, to protect the integrity of our test set to assess generalization, we explicitly moved this data into the training set, and assembled a new test set based on new data sources. In the end, the test set used to assess generalization of the final model are true holdout data, i.e. data that were never considered during the model development process, but only once the final model was available. We extended the methods section to clarify the composition of the training and test set, and updated the discussion section.

In considering this topic, we noted that the data points for L5 to L5 were extensively and fluidly used and explored throughout the model development process. As such, while the specific model parameterization presented in the paper was fit on only a sparse subset of L5 to L5 protocols, assessing the model generalization to other L5 to L5 protocols would be more correctly described as validation, as opposed to testing of generalization. Consequently, the section previously called "Model testing and predictions: optimized connection types" in the text has been renamed to "Model validation and predictions on the optimized connection types", and revised accordingly.

On the other hand, the authors discuss some data sets that were excluded because the experiments were from the wrong species, brain region or age, e.g. "we had experimental data for only one connection type and protocol featuring mostly apical synapses, and it was not from somatosensory cortex but rather the visual cortex." This seems to contradict an earlier part of the paragraph which states "we included data from rat visual cortex and from mouse somatosensory cortex" - clearly from two brain regions and two species. A more convincing argument needs to be made that the data sets were not selected based on the ability of the model to fit the data.

We agree with the reviewer's comment. It is important to show the readers that the experimental datasets were not "cherry picked" to artificially boost the generalization performance. Furthermore, a wider comparison of the model's results across brain regions and species constitutes a new interesting analysis that we are happy to add to the manuscript. For these reasons, we revised the test set to include all available data sources of paired recordings from rodent neocortex. In particular, we added: visual cortex experiments on L5 TTPC to L5 TTPC connections from Sjöström et al., 2001 (Supplementary Figure A.4); mouse barrel cortex experiments on L2/3 PC to L2/3 PC connections from Banerjee et al., 2014 (Figure 4); and visual cortex experiments on L2/3 PC to L2/3 PC connections from Zilberter et al., 2009 (Supplementary Figure A.6). To our knowledge there are no other in vitro paired recording experiments in the juvenile mouse or rat neocortex. Our test set is therefore now

comprehensive, and excludes any possibility for a biased selection of test data to boost generalization performance. The Results, Discussion and Methods sections have been revised, based on these new test set data sources and analyses. Furthermore, Table 1 and Table 2 were merged and updated to account for the new datasets, and Figure 4 was split to accommodate the new experiments on L2/3 PC to L2/3 PC connections.

3. The code is not provided. Neither the optimization code, nor the post-synaptic neuron models and synapse locations used for the simulations. The authors provide a URL for parameters, but this is not sufficient for reproducibility and it is unclear whether the rest of the code is part of the internal release and how to run the simulations. It is unclear until the very end of the manuscript if the entire column was simulated, or just the single post-synaptic neurons. If the latter, it seems that the purpose of the entire column was to determine connectivity. Line 678 indeed explains that the entire column was not simulated, but this needs to be stated much earlier in the manuscript.

We thank the reviewer for raising this important issue. We clarified in the manuscript that the circuit model was used as a constrained source of connected pairs of neurons (see beginning of Results section). We also remarked how sampling connections from a larger circuit model allowed us to transparently account for the variability of morphologies, synaptic model parameters and innervation profiles.

All the code used to run parameter optimizations and simulations of synaptic plasticity must be made available.

As requested by the reviewer, we have made all models and simulations used in this work publicly available on Zenodo (<https://zenodo.org/record/5654789#.YcCfCnvMIQ8>). We open sourced EModelRunner, a Python package to run the plasticity simulations (<https://github.com/BlueBrain/EModelRunner>). The optimization code is a minor extension of our previous “graupnerbrunelstdp” example already available in the open source BluePyOpt repository (<https://github.com/BlueBrain/BluePyOpt>), and it will be updated soon to reflect additions made for this work. We updated the Results and Methods sections to point to these new online resources.

Also, a table of parameters for equations provided in the text should be provided. The reader should not have to wade through the entire blue brain project trying to find model parameters.

All parameters for the synaptic plasticity model are included in Table 2. For all other relevant synaptic parameters, we updated the Supplementary Figures A.23 - A.27 to also report mean and SD, and reveal correlations between parameters for all the connection types considered in this work.

4. Calcium channel conductance and volume used in Equation 1 and described on lines 118-119 are unclear. The additional explanation in methods lines 572-573 does not help. The authors should state explicitly that there are no spines in this model. In addition, the explanation of spine volume seems circular since you determined spine volume from conductance, and then 2 sentences down in the methods stated that synaptic VDCC conductance used spine head dimensions. You mention a prescribed conductance distribution, which suggests it has been specified, but I don't see where the conductance distribution is specified. Equally important, what are the consequences for the plasticity rule of not making spine head volume proportional to calcium conductance? I see that correlation between spine head volume and AMPA conductance is finally explained in equation 51&52, but this does not include VDCC conductance.

Thanks to the reviewer for pointing out this obscure explanation. We have revised the text to clarify (lines 118-119 in original draft; lines 134-141 in the new draft). In brief, the synaptic conductance is taken as given and constant, as it is prescribed by constraints of the circuit model. The synaptic conductance is then used to determine the spine volume based on an established empirical relationship between these two variables. This procedure is important, because it normalizes calcium transients, as high-conductance synapses are assigned large spine volumes and vice versa, as is to be expected in the biological specimen. Spines are approximated as point processes, rather than as membrane compartments, for computational reasons. That is, although the spine membrane is not simulated, the spine volume is used to estimate the VDCC conductance (using surface density measurements reported in literature), and to compute the calcium concentration. A clarification of how VDCC is determined from volume was added after equations 51 & 52 (as per original draft).

5. The authors claim their calcium model is constrained by experimental data (lines 140-141), but do not reference calcium imaging publications that measure spine calcium during STDP or high frequency stimulation. Nor do they compare their calcium dynamics with other experimental measurements of calcium dynamics. The authors should provide direct experimental comparison in Figure 1 together with references used to constrain the calcium equation parameters.

We thank the reviewer for suggesting this revision. The calcium model was constrained by experimental measurements from Sabatini et al., 2002. We implemented the requested comparison as follows. We updated the Results and Methods sections in the manuscript and added a corresponding supplementary figure, showing good agreement between the model and the experimental data for both pre- and post-synaptic stimulation (Supplementary Figure A.1).

Minor comments.

In several places, including the discussion, the authors refer to phenomenological models as being insufficient for their purposes. But in fact this is a phenomenological model. The authors do not model diffusion, explicit buffers and pumps. Thus, it is a bit of a stretch to state (in the discussion) that "the model takes advantage of the full calcium time course", especially since the model requires an additional ad hoc, calcium integrator equation. Similarly, the bistable synaptic plasticity equation is phenomenological.

We agree with the reviewer that our model is also "phenomenological". We have revised the text to remove the problematic uses of the term.

line 229-230: It is unclear how being able to fit the synaptic plasticity data "supports our assumptions on the initial ratio of potentiated over depressed synapses, and the conductance and release probabilities of the depressed and potentiated states."

We apologize for the lack of clarity on this point. We refined the text in question to provide a more explicit explanation regarding the role of these parameters and the implications of their initialization on the achievable magnitudes of synaptic plasticity.

line 261: It is unclear how this is valid. Clearly, setting the ltd induction rate to 0 will block LTD, but it is unclear how this is comparable to blocking presynaptic NMDAR. Modifying the pre-synaptic equations would be more appropriate.

Rodriguez-Moreno and Paulsen, 2008 conclude that blocking presynaptic NMDARs abolishes LTD for this connection type, as has been reported for L5-L5 PC connections by Sjöström et

al., 2003. Our simulations are devised to explicitly evaluate if blocking LTD in our in silico model ($\gamma = 0$) leads to outcomes that are in quantitative agreement with the in vitro results. We found this to be the case (i.e. not just LTP, same magnitude of LTP), suggesting that the model can generalize to this connection type for more than one protocol. We revised the text to better reflect the above rationale.

line 318: The authors actually present data on plasticity that depends on pre-syn NMDA receptors, which the authors model cannot implement exactly. This data shows that the plasticity mechanisms are not shared and contradicts the authors claims.

The reviewer is correct in pointing out the existence of connection-type specific plasticity mechanisms between the PC types of the neocortex. This apparently contradicts our statement in question (line 318), due to its imprecise wording. We have revised it to clarify our position, which is as follows. While there is known diversity and connection-type specificity in pre- and post-synaptic induction mechanisms of LTD/LTP between PCs in the neocortex, the core role of postsynaptic NMDARs, VDCCs and postsynaptic calcium is broadly shared, and is well established. In this study, we hypothesized that postsynaptic calcium diversity is sufficient to explain connection-type specificity of *activity-dependent* plasticity outcomes between PCs in the neocortex. We fit a phenomenological model of plasticity driven by postsynaptic calcium on data from L5-L5 PCs (and others), and achieved good agreement, even though plasticity at this connection type is known to depend on presynaptic NMDARs for LTD (Sjöström et al., 2003). It follows that this complexity is accounted for phenomenologically in our model, suggesting that postsynaptic calcium dynamics are the principal component determining plasticity outcomes for the experiments and protocols considered here, regardless of all the details of the pre- and post-synaptic molecular machinery involved.

Line 481 and Eqn 15-16: Since this is describing the synaptic response to vesicle release, and you have distinct equations for the pre-synaptic side, why is N_{sites} part of this equation?

This is a pooled equation for all release sites at one synapse. Our formalism is designed to preserve the EPSP amplitude under changes in N_{sites} , and so it must appear in Eqns 15-16 and 8-9 as a normalizer. This approach allows N_{sites} to be parameterized independently (e.g. without affecting EPSP amplitude) to match experimental data on EPSP CVs, as described in Barros-Zulaica et al., 2019.

lines 499-516 are very confusing. 1st, why provide and explain an equation you then state you are not using. 2nd, at what values is it undefined? Doesn't that depend on Ca , M , p_m and p_{ca} , which I couldn't find in the manuscript? Why not use the GHK flux equation which does not normalize to all other currents, and which is well defined (using L'Hospital's Rule).

We thank the reviewer for raising this issue. The purpose of the approximation in lieu of the GHK flux equation is that $I_{Ca_NMDA} = P_f * I_{NMDA}$ is not a good approximation of the calcium current around the receptor reversal potential, where the net current is zero, although the calcium component is far from it. We revised the text to clarify and provide a more explicit argument.

This statement is unclear: "We assigned the initial state of the synapse to the potentiated ($\phi_0 = 1$) or depressed ($\phi_0 = 0$) state at random, using the individual synapse release probability, U_{SE} ". Please provide additional information.

As we are assuming a bimodal model, synapses are initialized in one of their stable states (either potentiated or depressed). To make this initialization consistent with the other model parameters, we decided to use the release probability, prescribed in the circuit model, also as the probability of being potentiated or depressed. This mechanism homogenizes synaptic transmission and plasticity parameterizations, preventing unreasonable initializations to take place (i.e fully potentiated connection, but unreliable due to a very small release probability). We revised the text to make this argument more explicit in the Methods section.

Eqn 36,37 are not complete. The term C_{pre} is not defined. It seems this is left over from the previously published model. Given that there are explicitly equations for vesicle release (lacking in the previously published model), this term should not be needed.

We apologize for the misunderstanding. C_{pre} and C_{post} are not left overs from the previous model. They are calibrated for each synapse by running a short simulation. In other words, the plasticity thresholds are normalized to the expected calcium transients at each synapse, allowing us to specifically parameterize each of them using the same " $a_{(i,j)}$ " scaling factors. We updated the Methods section to clarify this matter.

Eqn 38-42 are incomplete. g_{max} depends on g_d and g_p , but g_d and g_p depend on initial $g_{max,0}$, which is not defined in this section.

Thanks, we updated the variable names and highlighted those predefined in the circuit model.

Eqn 45-46 have a term x in them which is not defined, possibly that should be γ .

Correct, we fixed it.

line 673: I don't know what hot-fixed means.

Thanks. We removed this colloquial term from the text.

Figures A.2-A.11 have no y axis labels and none of figure captions explain the graphs' meaning.

We updated the captions and axis labels to clarify the meaning of these plots. In brief, they show the distributions of individual experiment outcomes for each protocol and connection type considered.

Reviewer #3 (Remarks to the Author)

The manuscript presents a model of long-term synaptic plasticity based on the dynamics of postsynaptic calcium. The novelty of this approach is that a single calcium-based mechanism is sufficient to reproduce experimental data from a number of synapses between different cortical cell types, with observed differences arising from other properties of the postsynaptic cells such as overall morphology and synapse location. The model combines detailed biophysical modelling of voltages and calcium currents with phenomenological elements to account for the integration of calcium signals on relatively long timescales and a bistable synaptic efficacy model introduced in Graupner and Brunel (2012) to allow for plasticity. The claims are novel in that a single model that can predict both depression and potentiation between detailed models of different cortical cell types, and potentially at different (in vivo) calcium concentrations, did not previously exist and the paper is likely to be of interest to the community. The stated intent of the paper is to provide a null model for cortical excitatory plasticity against which other models and data can be judged and I believe that this is

achieved; the results are likely to be a useful comparison to for future plasticity models with additional details or in different brain regions.

The models and methods of analysis are well described and build on a number of techniques described in previous papers from the Blue Brain Project. The implementation of the plasticity model is likely to be reproducible for single neurons by independent researchers, but the level of computational power required to reproduce the full set of results in the paper would be prohibitive for many.

We thank the reviewer for this thoughtful comment. While it is true that the full optimization procedure is computationally expensive, using the model would not require refitting in most cases. Furthermore, a local optimization around the best solution provided in this work would require far less simulations than our initial search. We made publicly available all connections in this work, together with all the required software to run the simulations and the analyses on regular desktop computers. As part of this process, we also noticed that common desktop CPUs perform particularly well, due to the higher clock frequency with respect to server CPUs. We revised the text to address the computational limitations of the study, together with a few suggestions on how to reduce computational cost, if needed.

I have a couple of more detailed comments:

Line 173 (etc): The correlation of (plasticity in) synaptic conductance and release probability is a strong assumption and is not entirely supported by the cited experimental literature, which only explicitly finds this relationship in hippocampal synapses. It is not unreasonable to make this assumption, but it does restrict the range of synaptic dynamics that can emerge, particularly given the importance of stochasticity in synaptic transmission and ongoing plasticity.

Thanks to the reviewer for raising this issue. We performed a new analysis of the locus of expression of synaptic plasticity at the whole connection level, following the methods in Sjöström et al., 2007. This analysis allowed the direct comparison of experimental results for visual cortical connections with our in silico model. Interestingly, we found our model could reproduce the results on concurrent presynaptic LTD and postsynaptic LTP of whole connections, even though the individual synapses can only express either LTP or LTD. However, we did not observe the reported pre-synaptic only LTD. The manuscript has been revised to include this new analysis (Supplementary Figure A.5), and the discussion has been extended accordingly.

Lines 370, 537 (etc): The fact that the number of vesicle release sites is fixed in the model is another feature that will influence the statistics of potentiated and depressed synapses. The authors do discuss the reasons for focussing on AMPAR-mediated plasticity as the determinant of EPSP amplitude in the Discussion, but it might also be interesting to comment on the implications of relaxing this restriction.

The reviewer raises a very interesting discussion point. In this work we do not consider active zone plasticity, as data on the relative contribution of release probability and release sites to presynaptic plasticity is still sparse. Furthermore, release probability has been repeatedly shown to be implicated in synaptic plasticity and is often considered the main target of presynaptic LTP/LTD. However, reports on plasticity of nanomodules (i.e. a tightly coupled organization of release sites and receptors) are indeed challenging this simpler view. We added a paragraph in the Discussion to address this interesting matter, together with the data requirements to account for it in the model.

Eqs 1, 13, 44: The symbol gamma is used to represent three different constants, it might be clearer to use different symbols.

Thanks for the suggestion, we thoroughly revised the equations and symbols to address these issues and kept gamma for the LTP/LTD rates in Graupner model.

Line 126: Typo 'vesicles'.

Thanks. Fixed: changed 'vesicles' (plural) to 'vesicle' (singular).

REVIEWERS' COMMENTS

Reviewer #1 (Remarks to the Author):

I appreciate the changes made by the authors. The manuscript has considerably improved following the revisions, can be used by other researchers in the field (as the relevant code is now available), and provides insights regarding the mechanisms for the emergence of the different plasticity protocols/predictions for new plasticity protocols that can be experimentally verified in the future. It should be noted that with the inclusion of the statistical analysis, the generalization of their results is now restricted. Yet, this model can be used as a starting point for these connections/areas that the model does not currently reproduce. Overall, I support this work for publication in Nat. Commun.

Minor comment:

Lines 325-326: Please consider rephrasing, as written it is confusing with how this is explained at lines 327-329.

Reviewer #2 (Remarks to the Author):

All my comments have been addressed. The additional figures and code are greatly appreciated. However, the additional comparison for synaptic plasticity in visual cortex (e.g. Fig A.5) shows major discrepancies. This suggests a strong region dependence (unless I missed that some of the data is from auditory cortex). Thus, I suggest changing "neocortex" to "somatosensory cortex" in title and elsewhere.

Fig 3, 7, etc. Please replace all rainbow/heat map color scales with color-blind friendly color scales

Reviewer #3 (Remarks to the Author):

The authors have thoroughly addressed my comments and I am happy with the revised manuscript.

Point-by-point response to the reviewers' comments

Reviewer #1 (Remarks to the Author)

I appreciate the changes made by the authors. The manuscript has considerably improved following the revisions, can be used by other researchers in the field (as the relevant code is now available), and provides insights regarding the mechanisms for the emergence of the different plasticity protocols/predictions for new plasticity protocols that can be experimentally verified in the future. It should be noted that with the inclusion of the statistical analysis, the generalization of their results is now restricted. Yet, this model can be used as a starting point for these connections/areas that the model does not currently reproduce. Overall, I support this work for publication in Nat. Commun.

Minor comment:

Lines 325-326: Please consider rephrasing, as written it is confusing with how this is explained at lines 327-329.

We thank the reviewer for this suggestion, and have revised the text for clarity.

Reviewer #2 (Remarks to the Author)

All my comments have been addressed. The additional figures and code are greatly appreciated. However, the additional comparison for synaptic plasticity in visual cortex (e.g. Fig A.5) shows major discrepancies. This suggests a strong region dependence (unless I missed that some of the data is from auditory cortex). Thus, I suggest changing "neocortex" to "somatosensory cortex" in title and elsewhere.

Thanks to the reviewer for pointing this out. We have revised the main text, and added an additional paragraph in the discussion to clarify our findings regarding the generalization across neocortical regions. Furthermore, we have revised the title to more correctly reflect what was possible to show in this study. We have opted to preserve *neocortex* in the title, abstract, and in a few places in the text to reflect the fact that our work does consider other cortical regions besides somatosensory cortex when evaluating generalization. Given the lack of other region specific cortical microcircuit models and the sparsity of available experimental data, region-to-region generalization could not be conclusively evaluated. Nonetheless, our results suggest the applicability of the modeling principles to other cortical regions. For example, distance dependent plasticity in the visual cortex could be reproduced in our model.

Fig 3, 7, etc. Please replace all rainbow/heat map color scales with color-blind friendly color scales.

We thank the reviewer for this attentive comment. The colormap used for the plasticity heat maps ("turbo") is actually already a color-blind friendly color scale. Specifically, it is a modern take by Google AI on the commonly used "jet" rainbow colormap, which is designed to solve some of the major shortcomings of the latter, including color blindness ambiguity (see <https://ai.googleblog.com/2019/08/turbo-improved-rainbow-colormap-for.html>). The authors present in the cited post the results of extensive testing of the "turbo" colormap using the Color BLindness Simulator (CoBlIS), showing indeed good performance in the vast majority of tested color blindness conditions.

Reviewer #3 (Remarks to the Author)

The authors have thoroughly addressed my comments and I am happy with the revised manuscript.

Thanks to the reviewer for their positive feedback.